# Contrasting impacts of humidity on the ozonolysis of monoterpenes: insights into the multi-generation chemical mechanism

Shan Zhang, Lin Du*, Zhaomin Yang, Narcisse Tsona Tchinda, Jianlong Li, Kun Li*

Environment Research Institute, Shandong University, Qingdao 266237, China.

*Correspondence to*: Lin Du (lindu@sdu.edu.cn) and Kun Li (kun.li@sdu.edu.cn)

**Abstract.** Secondary organic aerosol (SOA) formed from the ozonolysis of biogenic monoterpenes is a major source of atmospheric organic aerosol. It has been previously found that relative humidity (RH) can influence the SOA formation from some monoterpenes, yet most studies only observed the increase or decrease in SOA yield without further explanations of molecular-level mechanisms. In this study, we chose two structurally different monoterpenes (limonene with an endocyclic double bond and an exocyclic double bond, $\Delta^3$-carene with only an endocyclic double bond) to investigate the effect of RH in a set of oxidation flow reactor experiments. We find contrasting impacts of RH on the SOA formation: limonene SOA yield increases by ~100% as RH increases, while there is a slight decrease in $\Delta^3$-carene SOA yield. Although the complex processes in the particle phase may play a role, we primarily attribute it to the water-influenced reactions after ozone attack on the exocyclic double bond of limonene, which leads to the increment of lower volatile organic compounds under high RH condition. However, as $\Delta^3$-carene only has an endocyclic double bond, it cannot undergo such reactions. This hypothesis is further supported by the SOA yield enhancement of β-caryophyllene, a sesquiterpene that also has an exocyclic double bond. These results greatly improve our understanding of how water vapor influences the ozonolysis of biogenic organic compounds and subsequent SOA formation processes.

## 1 Introduction

Secondary organic aerosol (SOA), as an important type of ambient fine particulate matter (PM$_{2.5}$: aerosols with aerodynamic diameter $\leq 2.5$ μm) (Guo et al., 2014; Huang et al., 2014), has caused a series of negative impacts on human health (Pye et al., 2021), air quality (Zhang et al., 2016) and global climate (Levy et al., 2013). SOA produced from the oxidation of biogenic volatile organic compounds (BVOCs) is a major component of SOA in heavy forest regions during summer (Sindelarova et al., 2014; Ahmadov et al., 2012), and contributes by a large fraction (~40%-80%) to global OA budget (Cholakian et al.,

2019).

Monoterpenes, mostly emitted from coniferous trees, account for ~11% in total BVOCs

(Sindelarova et al., 2014; Kanakidou et al., 2005). Limonene is one of the most abundant monoterpenes,
with the annual emission budget of 11.4 Tg yr$^{-1}$ (Guenther et al., 2012). Apart from the biogenic source,
limonene can also be released from the indoor emission, mainly from essential oils (Ravichandran et al.,
2018; De Matos et al., 2019; Mot et al., 2022). Limonene has an endocyclic double bond and an exocyclic
double bond, and is thus more reactive than other monoterpenes towards oxidants such as ozone ($O_3$),
hydroxyl radical (OH), and nitrate radical ($NO_3$) (Chen and Hopke, 2010; Atkinson and Arey, 2003). $\Delta^3$-
carene is another kind of monoterpene that dominates the monoterpene emission from Scots pine (Bäck
et al., 2012). Different from limonene, $\Delta^3$-carene contains only one endocyclic double bond, which is
similar to most other monoterpenes.

Ozonolysis is an important reaction pathway for limonene and $\Delta^3$-carene. Although reactions with

OH and $NO_3$ are faster than that with $O_3$ for both two monoterpenes (Atkinson, 1991; Khamaganov and
Hites, 2001; Chen et al., 2015; Shaw et al., 2018), the atmospheric concentration of the latter
monoterpene is much higher than that of the former (Sbai and Farida, 2019). The contributions of $O_3$-
reactions with limonene and $\Delta^3$-carene to tropospheric degradation are 47% and 24%, respectively, in
the daytime (Ziemann and Atkinson, 2012). In pristine areas where $NO_3$ concentration is very low,
ozonolysis is also the dominant fate for limonene and $\Delta^3$-carene in the nighttime. In addition, it has been
previously found that the ozonolysis of monoterpenes can produce more extremely low volatility
products than OH-initiated oxidation, which contributes by a large fraction to the SOA production
(Jokinen et al., 2015). For either limonene or $\Delta^3$-carene, the first step for ozonolysis is attacking on the
endocyclic double bond to form two types of stabilized Criegee intermediates (sCI) with low energy (Fig.
S1) (Drozd and Donahue, 2011; Chen et al., 2019). The sCI will then trigger a series of chemical reactions,
like isomerization, decomposition and addition reactions. Correspondingly, the major components in $\Delta^3$-
carene SOA are caric acid, OH-caronic acid, and caronic acid (Ma et al., 2009; Thomsen et al., 2021),
while the major components from limonene SOA are limonaldehyde, keto-limonon aldehyde, limononic
acid and keto-limononic acid (Pathak et al., 2012; Wang and Wang, 2021).

Water is ubiquitous in the atmosphere and can affect the formation mechanism of SOA and its

relevant physical and chemical properties (Sun et al., 2013). A number of field measurements have shown
that the average molecular weight of the water/organic phase and activity coefficient of condensed
organics would be changed due to the change of relative humidity (RH) (Seinfeld et al., 2001; Li et al.,
2020). In addition, several laboratory studies have demonstrated that RH can influence the ozonolysis of
monoterpenes in different ways. Most of those studies have reported either an inhibitory effect or a
negligible effect of high RH on the particle formation (Bonn and Moortgat, 2002; Fick et al., 2002; Zhao
et al., 2021; Ye et al., 2018). Nevertheless, few other studies found that high RH can promote SOA
formation from the ozonolysis of limonene (Yu et al., 2011; Gong et al., 2018; Xu et al., 2021), but the
reason of this promotion effect remains unclear.

To fully examine the effects of water on SOA formation from the ozonolysis of monoterpenes,

especially the related chemical processes, we used an oxidation flow reactor (OFR) to investigate the
ozonolysis of limonene and $\Delta^3$-carene under different RH conditions in this study. An ultra-high
performance liquid chromatography with a quadrupole time-of-flight mass spectrometer (UPLC-Q-TOF-
MS) was deployed to analyze the molecular chemical composition of the SOA, which provided insights
into the physical and chemical processes influenced by the water content. With these state-of-the-art
techniques, we proposed mechanisms that may explain the inhibitory or enhancing RH effects on SOA
formation for different monoterpenes.
**2 Experimental methods**
**2.1 Oxidation flow reactor experiments**

A series of dark ozonolysis experiments of limonene and $\Delta^3$-carene were conducted in a custom-

made oxidation flow reactor (OFR). The OFR is a 602 mm long stainless cylinder with a volume of 2.5
L (Fig. S2) (Liu et al., 2019; Liu et al., 2014). A zero-air generator (XHZ2000B, Xianhe, China) was
used to generate dry clean air as the carrier gas for the OFR. As shown in Fig. S2, there are four gas paths
upstream of the OFR: the first path is the precursor gas channel through which monoterpenes are injected
via a syringe pump (ISPLab 01, Shenchen, China); the second path is for the flow of 300 sccm dry zero
air passing through a mercury lamp ($\lambda = 185$ nm) to generate $O_3$; the third path is connected to a water
bubbler to generate wet air; the fourth path is the extra dry zero air entering the OFR. The RH in the OFR
was controlled by adjusting the ratio of the wet and dry zero air flows. A water recycle system was
equipped to keep the temperature (T) around at 298 K. The total flow was 0.9 L min$^{-1}$, resulting in an
average residence time of 167 s. The RH and T in the OFR were monitored by a T/RH Sensor (HM40,
VAISALA, Finland). The concentration of ozone and the consumption of the precursor gas were
measured with an ozone monitor (Model 106L, 2B Technologies, USA) and a gas chromatography with
flame ionization detector (GC-FID 7890B, Agilent Technologies, USA), respectively. The GC was
equipped with a DB-624 column (30 m × 0.32 mm, 1.8 µm film thickness) whose temperature was set
to ramp from 100 °C to 180 °C at a rate of 20 °C min$^{-1}$, and then held at 180 °C for 2 min. Before each
experiment, $O_3$ was introduced into the OFR to clean it until the background aerosol mass concentration
reached < 1 µg m$^{-3}$.
The experimental conditions are shown in Table 1. In these OFR experiments, the precursor
(limonene or $\Delta^3$-carene) concentration was set to ~320-340 ppb. A high $O_3$ concentration of ~6 ppm was
used to realize an equivalent aging time of 0.41 day in the real atmosphere, assuming an average ambient
$O_3$ concentration of 28 ppb (Sbai and Farida, 2019) (see Section S1 for the calculation). Under such
conditions, most of the precursors were consumed, since the residence time was almost five and three
times of the half-life for limonene and $\Delta^3$-carene, respectively. Correspondingly, the $O_3$ consumption for
limonene and $\Delta3$-carene were ~250 ppb and ~100 ppb, respectively. A series of RH conditions ranging
from dry (1-2%) to 60% with a step of ~10% were used to investigate the effects of water content on
SOA production and composition (see Table 1). All materials used in the experiments have been
described in Section S2.
**Table 1.** Experimental conditions and results.

| Exp. | [Precursor] (ppb) | [O]$_3$ (ppm) | T (K) | RH (%) | N$_{(13.8-723.4 nm)}$[a] (cm$^{-3}$) | M$_{(13.8-723.4 nm)}$[b] (µg m$^{-3}$) | D$_{(mean)}$[c] (nm) | SOA yield (%) |
|---|---|---|---|---|---|---|---|---|
| | | | | limonene | | | | |
| 1 | 321±39 | 5.7 | 298 | 1–2 | 6.9×10$^5$ | 980.9 | 138.2 | 62.9 |
| 2 | 321±39 | 6.0 | 298 | 10±2 | 1.3×10$^6$ | 1377.5 | 126.8 | 88.4 |
| 3 | 321±39 | 5.9 | 298 | 20±2 | 9.0×10$^5$ | 1573.3 | 150.9 | 90.2 |
| 4 | 321±39 | 5.9 | 298 | 30±2 | 1.4×10$^6$ | 1573.3 | 128.9 | 100.9 |
| 5 | 321±39 | 6.0 | 298 | 40±2 | 1.7×10$^6$ | 2051.4 | 130. 7 | 131.6 |
| 6 | 321±39 | 5.5 | 298 | 50±2 | 1.5×10$^6$ | 1962.7 | 137.8 | 125.9 |

| 7 | 321±39 | 5.5 | 298 | 60±2 | $1.5\times10^6$ | 2211.1 | 139.0 | 141.8 |
|---|---|---|---|---|---|---|---|---|
| | | | | $\Delta^3$-carene | | | | |
| 8 | 341±28 | 6.1 | 298 | 1–2 | $9.5\times10^4$ | 346.0 | 195.8 | 19.4 |
| 9 | 341±28 | 6.4 | 298 | 10±2 | $1.4\times10^5$ | 300.3 | 163.4 | 16.8 |
| 10 | 341±28 | 6.4 | 298 | 20±2 | $9.4\times10^4$ | 244.9 | 176.9 | 13.7 |
| 11 | 341±28 | 6.0 | 298 | 30±2 | $5.9\times10^4$ | 241.2 | 205.1 | 13.5 |
| 12 | 341±28 | 6.3 | 298 | 40±2 | $4.6\times10^4$ | 205.8 | 203.2 | 11.5 |
| 13 | 341±28 | 6.3 | 298 | 50±2 | $6.8\times10^4$ | 196.7 | 180.7 | 11.0 |
| 14 | 341±28 | 6.3 | 298 | 60±2 | $5.6\times10^4$ | 198.5 | 190.2 | 11.1 |

[a] $N_{(14.1-735\ nm)}$ means the total particle number concentration from size 13.8 nm to 723.4 nm. [b] $M_{(13.8-723.4\ nm)}$ means the total particle mass concentration from size13.8 nm to 723.4 nm. [c] $D_{(mean)}$ means the particle mean diameter.

**2.2 SOA particle analysis**

**2.2.1 SOA yield**

The SOA particle size distribution was measured with a scanning mobility particle sizer (SMPS), which consists of a differential mobility analyzer (DMA) (model 3082, TSI Inc., USA) and a condensation particle counter (CPC) (model 3776, TSI Inc., USA). The samples were measured by SMPS every 5 minutes with a sampling flow and a sheath flow of 0.3 L min$^{-1}$ and 3 L min$^{-1}$, respectively. The SOA mass concentration was calculated from the volume concentration measured with SMPS and the aerosol density, which was estimated to be 1.25 cm$^{-3}$ for limonene- and 1.09 g cm$^{-3}$ for $\Delta^3$-carene-SOA (Thomsen et al., 2021; Watne et al., 2017).

The SOA yield (Y) for individual organic gas can be calculated as:

$$Y=\frac{\Delta M}{\Delta HC}$$

Where $\Delta M$ is the total mass concentration of SOA, $\Delta HC$ is the mass concentration of reacted precursor (Ng et al., 2007; Odum et al., 1996).

**2.2.2 Ultra-high performance liquid chromatography quadrupole time-of-flight mass spectrometry analysis**

An ultra-high performance liquid chromatography (UPLC, UltiMate 3000, Thermo Scientific) coupled with a quadrupole time-of-flight mass spectrometry (Q-TOFMS, Bruker Impact HD) was used

to analyze the molecular-level chemical composition of SOA. First, the SOA particles were collected on
the PTFE filters (47 mm diameter, 0.22 μm pore size, Jinteng, China). Next, these filters were dissolved
and extracted by 5 mL methanol for two times. Extracts were then filtered through PTFE syringe filters
(0.22 μm pore size), and were concentrated to near dryness by nitrogen-blowing. At last, the samples
were redissolved in a 200 μL solution with 0.1% (v/v) formic acid in 50:50 methanol/ultrapure water
mixture.
The parameters of LC-MS were set as follows: capillary voltage 4000 V, nebulizer pressure 0.4 bar,
dry heater temperature 200℃, end plate voltage −500 V, and flow of dry gas 4 L min$^{-1}$. A $C_{18}$ column
(100 Å, 3 mm particle size, 2.1 mm×50 mm, Waters, USA) was used with a column temperature of 35℃.
The mobile phase was 0.1 % formic acid in methanol (A) and 0.1 % formic acid in ultra-high purity
water (B) with a flow of 200 μL min$^{-1}$. The injection volume was 5 μL. The MS was operated in negative
ion mode, and the detection molecular weight range was from m/z 50 to 1500. The temperature ramp
program was: 0–3min with 0%–3% phase B, 3–25min with 3%–50% phase B, 25–43min with 50%–90%
phase B, 43–48 min with 90%–3% phase B, 48–60min with 3% phase B.
**3 Results and discussion**
**3.1 SOA production under different RH conditions**
SOA formation of a representative experiment is shown in Fig. S3. It is found that the formed SOA
are mainly in the size range of 60-200 nm, and the number concentration and mass concentration are
relatively stable during the course of the OFR experiment. SOA formation from limonene and $\Delta^3$-carene
in terms of particle number concentration, particle mass concentration, and SOA yield as a function of
RH are illustrated in Fig. 1a-c. We find that all the above-mentioned 3 parameters of limonene-SOA
increase with the increasing RH. The increment of particle mass concentration and SOA yield from the
ozonolysis of limonene is ~100% higher at wet (60% RH) than at dry conditions. In contrast, SOA
formation from $\Delta^3$-carene is suppressed by ~40% under high RH. The distinct effects of RH on SOA
formation from the ozonolysis of limonene and $\Delta^3$-carene found in this study agree with most previous
studies (Yu et al., 2011; Jonsson et al., 2006b; Bonn et al., 2002; Gong and Chen, 2021; Li et al., 2019b).
As shown in Table 2, Yu et al. (2011) reported a positive correlation between SOA production and RH
for the ozonolysis of limonene in the chamber experiments without OH scavenger. Their experimental

condition is similar to that in our study regarding the absence of OH scavenger and, thus, similar results were observed. However, in the presence of OH scavenger, results are quite different. Jonsson et al. (2006) observed a similar enhancement effect of high RH on SOA production with 2-butanol as the OH scavenger, while Bonn et al. (2002) found a negligible or suppressive effect with cyclohexane as the OH scavenger. It should be noted that the OH scavenger not only has the ability to scavenge OH but also produces additional products which may influence the reactions of target precursors. For example, there is no difference between 2-butanol and cyclohexane in the scavenging ability of OH radical, though 2-butanol will produce more $HO_2\cdot$ than cyclohexane and, consequently, $R\cdot$ will react with $HO_2\cdot$ to produce more hydroxyl acids and hydroxyl per-acid products, most of which have low volatility and, thus high partitioning into the particle phase. According to previous studies, the influence of different OH scavengers can vary (Jonsson et al., 2008). This may explain the different findings with and without OH scavenger for limonene-SOA. With regard to $\Delta^3$-carene, similar results are found in the absence of OH scavenger, namely, high RH has negligible or slightly suppressive effect on SOA production (Bonn et al., 2002; Fick et al., 2002). Same as limonene, the presence of OH scavenger and its different chemical nature can explain the different results found previously (Jonsson et al., 2006a; Bonn et al., 2002).

The enhancement in limonene-SOA production under high RH can be due to several reasons from either physical or chemical processes. First, the hygroscopic growth of the particles (i.e., absorption of water content) can lead to higher mass concentration under higher RH, but the enhancement should be at most ~30% as the growth factor (GF, the ratio of wet and dry diameter: $D_{wet/Ddry}$) of limonene-SOA is $\leq 1.1$ (Varutbangkul et al., 2006). However, we do not observe an obvious change in the mean diameter when comparing dry and wet conditions (Fig. 1d). In addition, hygroscopic growth should also occur for $\Delta^3$-carene SOA, but no obvious enhancement in particle mass is observed (Fig. 1a). Therefore, it is suggested that physical processes regarding hygroscopic growth play a minor role in the enhancement in limonene-SOA under high RH. As a consequence, we believe that chemical processes are likely the reason of the enhancement in limonene-SOA under high RH. Water can influence chemical processes in the gas phase or in the particle phase. Particle-phase reactions can promote the growth of small particles and, thus, mainly lead to larger particle sizes. As the observed SOA enhancement is mainly from high number concentration particles rather than the large size particles (Fig. 1b and 1d), it is likely that the water-participated gas-phase reactions are the most possible reasons for the limonene-SOA enhancement.

The reaction mechanism is analyzed below based on the mass spectra information on the SOA.

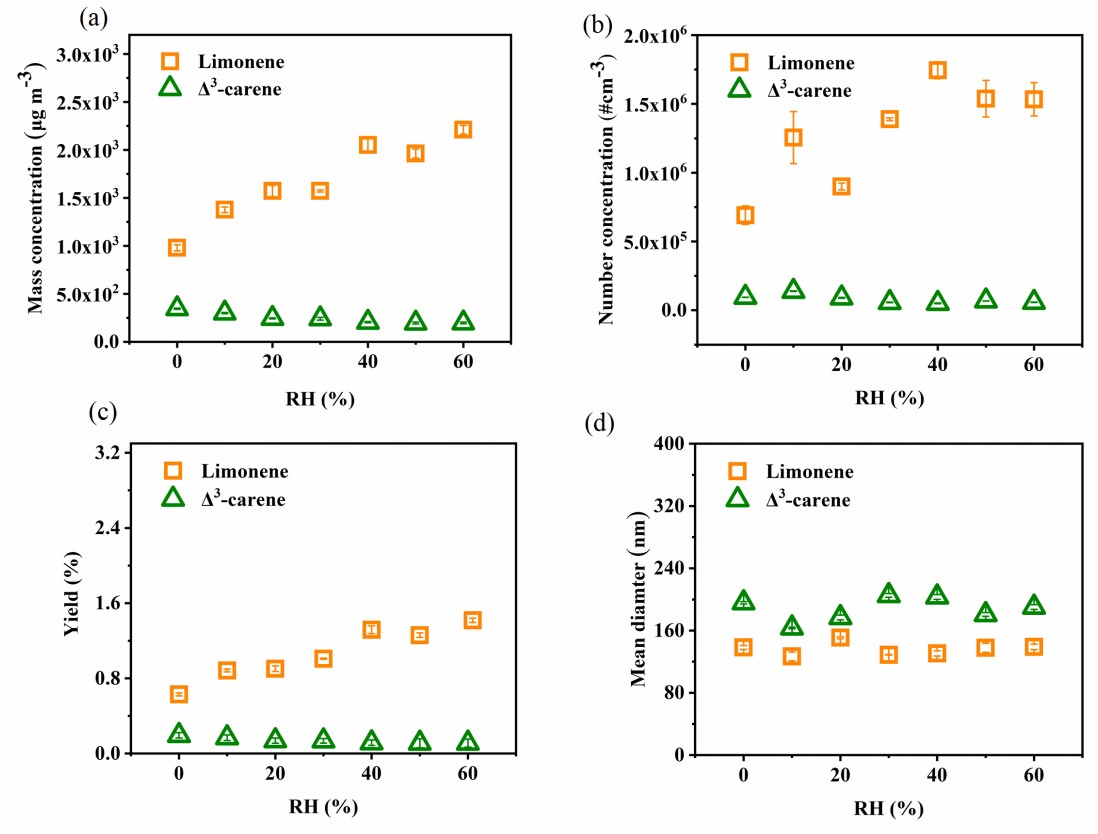


**Figure 1.** The effect of RH on the SOA formation: (a) number concentration, (b) mass concentration, (c)
SOA yield, (d) mean diameter.

**Table 2.** Comparison with previous studies on the effect of RH.

| Precursor | Precursor concentration (ppb) | O₃ concentration (ppb) | Reactor | OH scavenger | T (K) | RH (%) | SOA Mass Concentration (µg/m³) | SOA Yield (%) | Mᵃ | Nᵇ | Reference |
|---|---|---|---|---|---|---|---|---|---|---|---|
| limonene | 1000 | 1000 | flow reactor | cyclohexane | 295±2 | 0.02 and 32.5 | N.Mᶜ | N.Mᶜ | no effect | —ᶜ | Bonn et al. (2002) |
| | 320 | 100±5 | chamber | N.M.ᶜ | 296±2 | 18±2, 50±3 | 24; 58; 120 | 7.0 ± 0.7; 17.4±1.3; 53.4±1.9 | +ᵈ (7 times) | +ᵈ (8 times) | Yu et al. (2011) |
| | 15 and 30 | 430.9 | flow reactor | 2-butanol | 298±0.4 | < 2-85 and 82±2 | 2.7-10.5 and 62-229 | 6.8-26.4 and 77.4-285.7 | +ᵈ | +ᵈ | Jonsson et al. (2006) |
| | endocyclic (24.6) and exocyclic (15.2) | endocyclic (270) and exocyclic (12200) | flow reactor | 2-butanol | 298 | 10-50 | endocyclic (~11) and exocyclic (22-51) | endocyclic (~7.4) and exocyclic (23.8-55.3) | endocyclic (+ᵈ) and exocyclic (—ᶜ) | N.Mᶜ | Gong and Chen (2021) |
| | 1085 | 900±10 | flow reactor | none | 298 | 3-62 | 150; 200; 210 | N.M | +ᵈ | —ᶜ | Li et al. (2019) |

| | | | | | | | | | M[a] | N[b] | |
|---|---|---|---|---|---|---|---|---|---|---|---|
| Δ³-carene | 321±39 | 5786±203 | flow reactor | none | 298 | 0-60 | N.M[c] | N.M[c] | +[d] (2 times) | +[d] (3 times) | this study |
| | 1000 | (1000) | flow reactor | cyclohexane | 295±2 | 0.02 and 32.5 | 980.9-2211.1 | 62.9-141.8 | no effect | −[e] | Bonn et al. (2002) |
| | 14.2 and 29.4 | 2300 | flow reactor | 2-butanol | 298±0.4 | < 2-85 | 15.3-94; 19.8-116.7 | 0.78-3.8 and 2.1-10.1 and 19.8-116.7 | +[d] | +[d] | Jonsson et al. (2006) |
| | 1111 | 900±10 | flow reactor | none | 298 | 3-62 | 75; 80; 90 | N.M | −[e] | −[e] | Li et al. (2019) |
| | 341±28 | 6257±140 | flow reactor | none | 298 | 0-60 | 346.0-198.5 | 19.4-11.1 | no effect | no effect | this study |


[a] M means the change trend total particle mass concentration. [b] N means total particle number concentration. [c] N.M. means not mentioned. [d] Positive sign (+) means the mass or number concentration increases with RH. [e] Negative sign (−) means the mass or number concentration decreases with RH.

**3.2 Molecular analysis of SOA particles**

The UPLC/ESI-Q-TOF-MS was used to examine the SOA molecular composition under high and low RH conditions. As shown in Fig. 2a, the mass spectra of limonene-SOA are divided into four groups: monomeric group ($<$m/z 300), dimeric group (m/z 300-500), trimeric group (m/z 500-700), and tetrameric group (m/z 700-1000), corresponding to products containing one, two, three, and four oxygenated limonene units, respectively (Bateman et al., 2009). Most of the SOA molecules are monomers ($>$60%) (Fig. 2b) and dimers ($\sim$25%), while trimers and tetramers contribute to very small fractions ($<$10% and $\sim$3%) (Table S1). Correspondingly, the distribution of $\Delta^3$-carene-SOA can be divided into four groups (Fig. S4), comparable to that of limonene-SOA. Most of the SOA molecules are monomers ($\sim$70%) and dimers ($\sim$25%), while trimers and tetramers contribute to smaller proportions ($\sim$2% and $<$1%, respectively) (Table S2). Although the SOA mass concentration increases by $\sim$100% under high RH condition, the relative intensities of MS peaks do not significantly change with varying RH conditions. In other words, we did not observe an obvious change in the overall MS patterns, and the fractions of the four groups only slightly differed under different RH conditions, e.g., the fraction of monomers was 62% under dry condition and 66% under wet conditions. However, if we take a closer look, the intensities and contributions of specific peaks are quite different with varying RH. For example, the relative intensity of $C_{10}H_{16}O_2$, a possible first-generation product (Gong et al., 2018), decreases by $\sim$20% with increasing RH from dry to 60% (Table S3). This is likely due to the multi-generation reactions influenced by water vapor concentration, as discussed below with the proposed reaction mechanism of limonene ozonolysis.

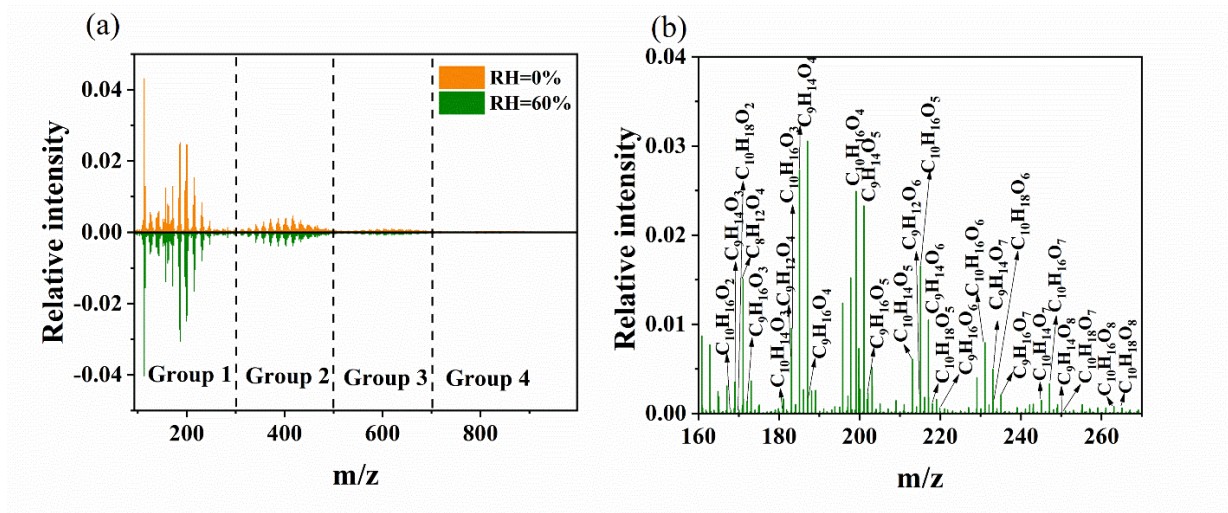

**Figure 2.** UPLC/ (−) ESI-Q-TOF-MS mass spectra of SOA from limonene ozonolysis. (a) MS under
high and low RH conditions; (b) the identification of monomers under high RH condition.
The proposed reaction mechanism of limonene ozonolysis is shown in Fig. 3 and Fig. 4. The initial
step in the reaction of $O_3$ with limonene is the attack of the endocyclic double bond to form $eCI_1$ and
$eCI_2$ (with branching ratios of 0.35 and 0.65, respectively). In the context of $eCI_1$, several complex
reactions occur, with the most dominant reaction being the generation of hydroxyl radicals (OH) and a
reaction pathway known as $sCI_1$. The $sCI_1$ pathway can proceed through three distinct reactions, as
depicted in Fig. 3. The first pathway is the reaction with $H_2O$, alcohol or carboxylic acid to form a
carboxylic acid species with hydroxyl, which would subsequently lose a molecule of water to form
limononaldehyde or lose a molecule of hydrogen peroxide to form limononic acid (Grosjean et al., 1992;
Li et al., 2019b). The second and third pathways involve reactions of $sCI_1$ with carboxylic acids and
carbonyls, respectively, leading to the formation of anhydrides and secondary ozonides. Additionally, the
generated OH radicals can react with limonene, giving rise to another alkyl radical, $C_{10}H_{17}O \cdot$. These
alkyl radicals react with $O_2$ and form peroxy radicals ($RO_2 \cdot$). The atmospheric fate of produced $RO_2 \cdot$ in
the absence of $NO_x$ includes the reaction with $RO_2 \cdot$ or $HO_2 \cdot$ (Atkinson and Arey, 2003) and the
unimolecular H shift. The $RO_2 \cdot + HO_2 \cdot$ route mainly form hydroperoxide (ROOH), and the minor fraction
is to form alcohols and carbonyls (Atkinson and Arey, 2003). The products of bimolecular reactions
between $RO_2 \cdot$ and $RO_2 \cdot$ are alcohols, carbonyls, alkoxy radicals, peroxides and ROOR dimers (Hammes
et al., 2019; Peng et al., 2019). The H shift of $RO_2 \cdot$ can form second-generation $R \cdot$ and trigger a main
generation channel of highly oxidized molecules (HOMs), i.e., $R \cdot$ would go through a process of repeated
oxygen addition and hydrogen-atom shift to form HOMs with high O/C ratios of > 0.7–0.8 (Molteni et
al., 2018; Bianchi et al., 2019).
In addition to the $eCI_1$ route, the $eCI_2$ pathway is also responsible for the generation of various
products (Fig. 4). Since the reaction of the hydroxyl radical (OH) attacking limonene is already depicted
in Fig. 3, our main emphasis in Fig. 4 is on the pathways involved in the generation of SCI. First, $sCI_2$
reacts with $H_2O$ and decomposes to limononaldehyde and $H_2O_2$. Additionally, $sCI_2$ could experience an
$O_2$ addition, $\cdot OH$ loss and isomerization to produce two types of $RO_2 \cdot$, which can undergo the similar
reactions as the $RO_2 \cdot$ formed from the $sCI_1$ route, and the major products are also shown in Fig. 4.
Since limonene and $\Delta^3$-carene both have an endocyclic double bond, the similar reactions as
mentioned above can occur for the ozonolysis of $\Delta^3$-carene (Fig. S5), and most corresponding formula
in Fig. S5 could be identified in Table S4. However, the reactivity of limonene towards $O_3$ is expected to
be higher owing to its exocyclic double bond. As shown in Fig. 4, the attack of $O_3$ to the exocyclic double
bond mainly leads to $sCI_3$ (highlighted in red) with the unpaired electrons outside the ring (Leungsakul
et al., 2005). $sCI_3$ can react with $H_2O$ to form a carbonyl called keto-limonene. It should be noted that
this reaction can occur not only for limonene, but also for all the products that retain the exocyclic double
bond. As a result, the compounds that are colored in blue in Fig. 3 and Fig. 4 can undergo further reactions
to generate products with an additional carbonyl (see the boxes in Fig. 3 and Fig. 4). Furthermore, their
molecular formula shown in Table S5 have been identified using the Q-TOF-MS. This mechanism can
well explain the decrease in the relative intensity of $C_{10}H_{16}O_2$ from high RH to low RH and the increase
in the relative intensity of $C_9H_{14}O_3$ from low RH to high RH (Table S3).

In such progress, we cannot rule out the possibility that relative humidity (RH) may influence the

generation of other free radicals (Ma et al., 2009), thereby impacting the formation of secondary organic
aerosols (SOA), such as, OH-radical reactions (Bonn et al., 2002; Fick et al., 2002). However, Molar OH
radical yields were reported as $0.65\pm0.10$ (Hantschke et al., 2021), $0.86\pm0.11$ (Aschmann et al., 2002)
and 0.56 to 0.59 (Wang et al., 2019) for $\Delta^3$-carene, while for limonene, the reported yields were $0.67\pm0.10$
(Aschmann et al., 2002) and $0.76\pm0.06$ (Herrmann et al., 2010). It seems that the OH radicals produced
from limonene and $\Delta^3$-carene are quite similar within the range of uncertainties. Therefore, the increased
ozone consumption by limonene seems primarily attributed to the presence of its exocyclic double bond.

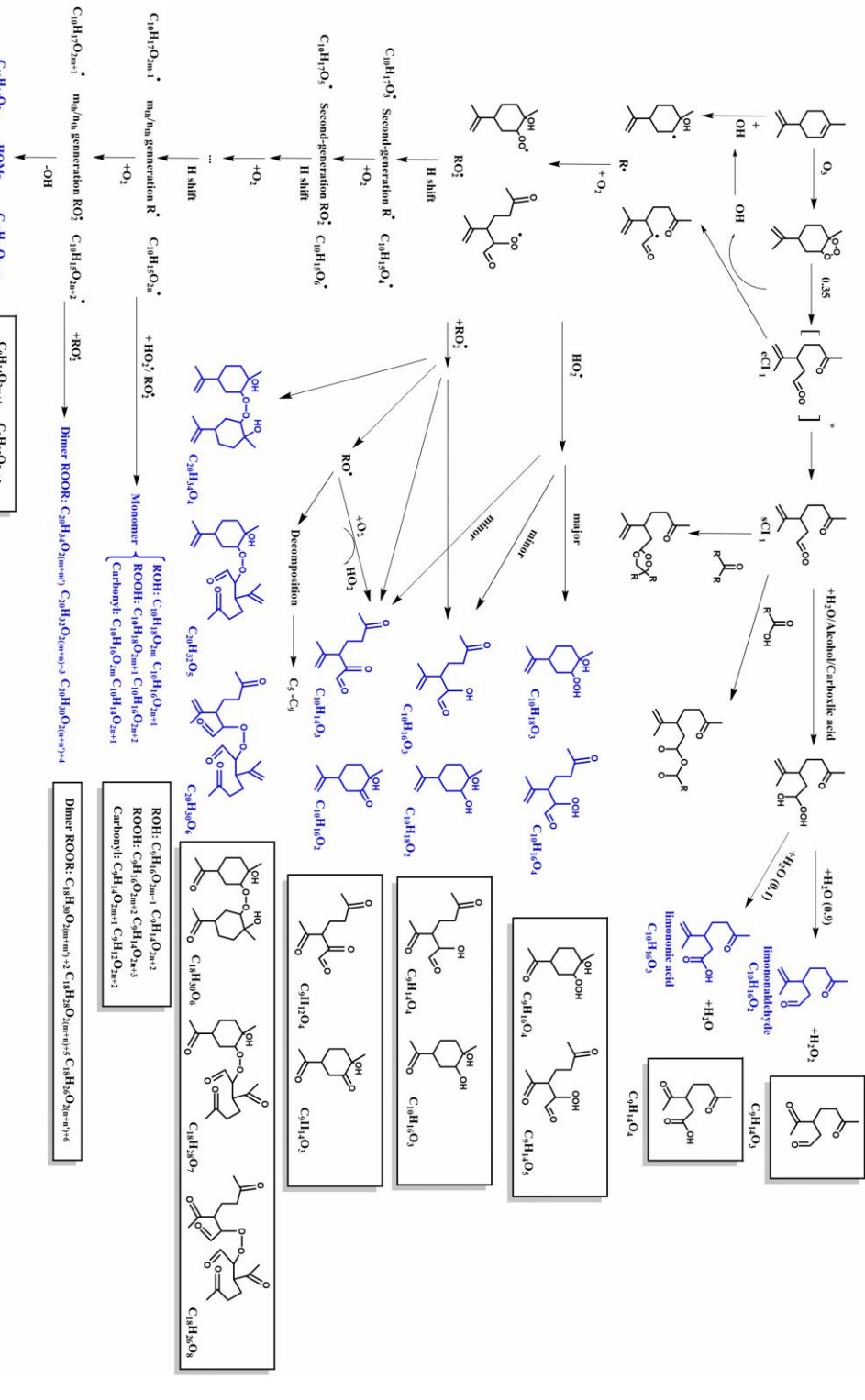


**Figure 3.** Proposed formation mechanism for SOA formation from eCI₁ oxidation under high RH. The compounds in blue and in boxes are identified using UPLC/ (−) ESI-Q-
TOF-MS.

**Figure 4.** Proposed formation mechanisms for SOA formation from eCI$_2$ and exocyclic double bond oxidation under high RH. The compounds in blue and in boxes are identified using UPLC/ (−) ESI-Q-TOF-MS.

**3.3 Processes leading to the increase or decrease in SOA formation**

Based on the results and mechanisms shown above, we present evidence that high humidity enhances limonene-SOA formation. First, the presence of water vapor enhances the formation of carbonyls from the reaction of exocyclic double bond, and the oligomerization of these carbonyls generates more dimers including hemiacetal (or acetal) formation and aldol condensation (Zhang et al., 2022; Kroll et al., 2005; Jang et al., 2003). As shown in Table S6, the intensity of dimers generating from multi-carbonyls under high RH is higher than that under low RH, and 54 out of the total 187 dimers were exclusively observed for limonene under high humidity conditions, contributing to a corresponding intensity of ~19% (Table S7). These dimers can be classified as low-volatile organic compounds (LVOC; $3\times10^{-4} < C_0 < 0.3$ μg m$^{-3}$) and extremely low-volatile organic compounds (ELVOCs; $C_0 < 3\times10^{-4}$ μg m$^{-3}$) (Fig. 5a), and thus promote the nucleation and new particle formation in different ways. This finding is similar to that from a previous study showing that high RH can promote dimer formation from the ozonolysis of α-pinene (Kristensen et al., 2014). Second, we find that high RH can also promote the formation of HOMs, although the mechanism remains unclear. As shown in Table S3, many HOMs proposed from the mechanism are detected under high RH condition but not detected under low RH condition, including both monomers and dimers. Many HOMs have low volatilities and, thus, can also promote new particle formation. Overall, the promoted dimer and HOM formation may greatly enhance the new particle number concentration under high RH condition (Fig. 6).

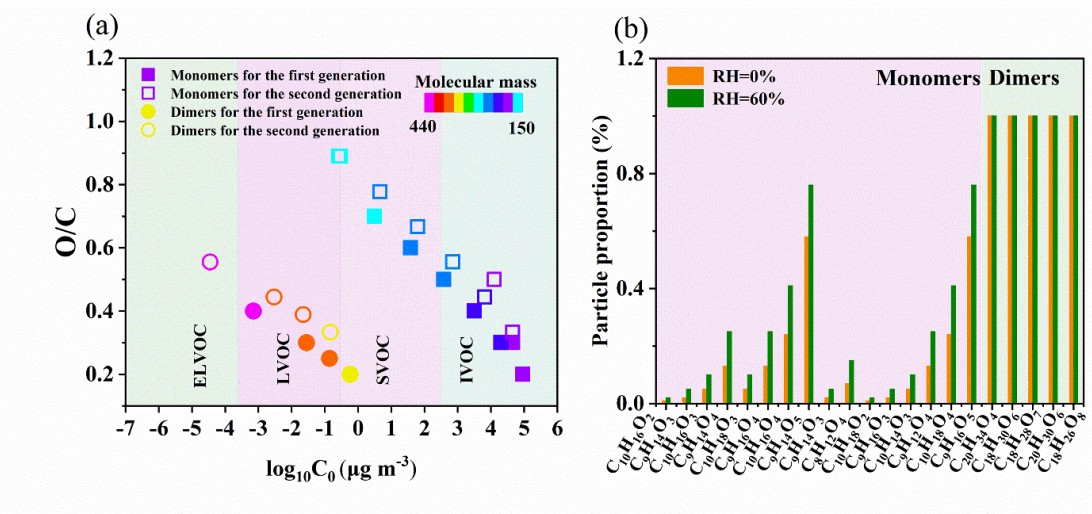


**Figure 5.** (a) Distribution of the limonene-SOA in the two-dimensional volatility basis set (2D-VBS)

space. (b) Partitioning coefficients of limonene monomers and dimers under low and high RH conditions.


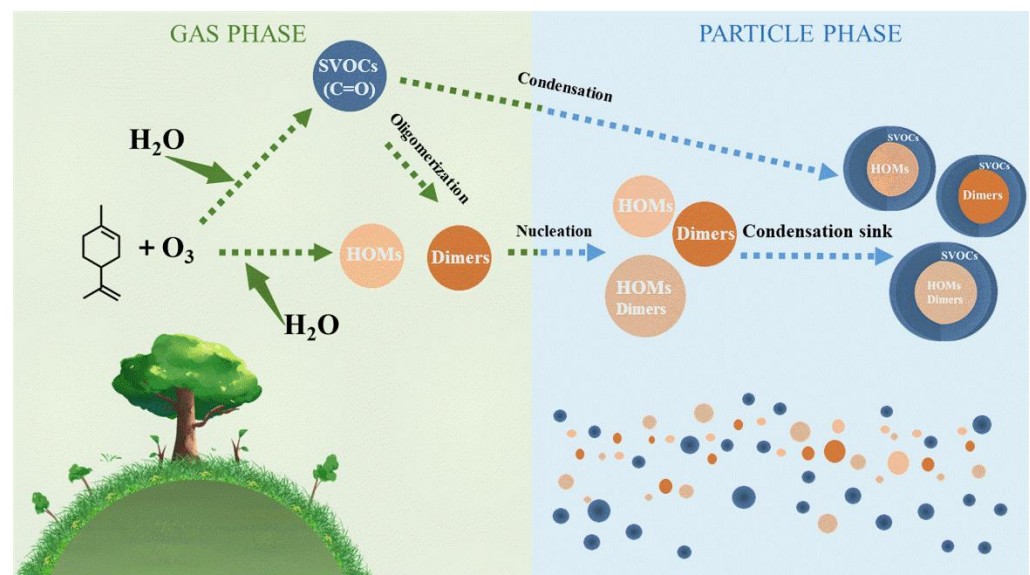


**Figure 6.** Schematic diagram of the possible mechanisms for the enhancement of limonene-SOA.


High particle number concentration generally provides more surface areas for semi-volatile organic

compounds (SVOCs; $0.3 < C_0 < 300$ μg m$^{-3}$) to condense on, which results in higher condensation sink

(CS). In the OFR, the fates of SVOCs include condensing on aerosol, getting lost on the wall, and reacting

with OH radicals to form functionalization and/or fragmentation products (Palm et al., 2016; Li et al.,

2019a). The promoted condensation by higher CS leads to a higher fraction of SVOCs getting into the

particle phase rather than getting lost on the wall or becoming smaller fragments staying in the gas phase,

and thus promoting SOA formation (Li et al., 2019a). Furthermore, the transformation from C-C double
bond to carbonyl shown in Fig. 3 and Fig. 4 decreases the volatility of molecules, which can largely
influence the gas-particle partitioning of the monomeric compounds (Fig. 5b). For example, the $C_0$ values
of $C_{10}H_{16}O_2$ and $C_{10}H_{16}O_3$ are 90701 and 19968 µg m$^{-3}$, corresponding to partitioning coefficients of
0.01 and 0.05, respectively (Fig. 5b and Table S3), with an SOA mass concentration of ~1000 µg m$^{-3}$
under dry condition. When they are converted to carbonyls $C_9H_{14}O_3$ and $C_9H_{14}O_4$, the values of $C_0$
become 45556 and 6479 µg m$^{-3}$, corresponding to partitioning coefficients of 0.02 and 0.13, respectively
(Fig. 5b and Table S3), with the same SOA loading. This enhancement in partitioning coefficient can
largely promote the condensation of SVOCs and, thus, enhance the SOA mass concentration. In addition,
the enhanced SOA formation can further influence the equilibrium, e.g., the partitioning coefficient of
$C_{10}H_{16}O_3$ increases from 0.05 to 0.10 when SOA mass concentration increases from ~1000 µg m$^{-3}$ under
dry condition to ~2000 µg m$^{-3}$ under wet condition (Fig. 5b and Table S3). The distribution of saturation
vapor pressure for monomers and dimers identified by MS has also been shown in Fig. 5a. As can be
seen from this figure, around 50% monomers are categorized as SVOCs, thus having the large fraction
in the particle phase when converting from dry to wet conditions. Overall, it is likely that the different
fate and partitioning of SVOCs largely enhance the amount of SVOCs in the particle phase (Fig. 6).
Concluding the analysis above, high humidity promotes the SOA formation from the ozonolysis of
limonene in two steps: nucleation of new particles and condensation of SVOCs on them (Fig. 6). While
our study highlights significant changes in gas-phase chemistry, we cannot exclude the possibility of
concurrent reactions occurring in the condensed phase. These two steps are closely related to the multi-
generation reactions of the exocyclic C=C bond, which are unlikely to happen for the ozonolysis of $\Delta^3$-
carene. Interestingly, Gong and Chen (2021) have found that high RH can inhibit the SOA formation
from the first-generation oxidation of limonene ozonolysis, but enhance the SOA formation from the
second-generation oxidation (Gong and Chen, 2021), their results agree well with the results and analysis
shown here. In contrast, Li et al. (2019b) found negligible change in dimers and HOMs in limonene-$O_3$
system when changing RH from 0 to 60%. The discrepancy is mainly attributed to the different
experimental conditions. The ozone exposure in this study is ~18 times higher than in Li et al. (2019b),
while the limonene concentration in this study is only ~30% of that in their study. These two conditions
both favor the multi-generation reactions occurred at the exocyclic double bond of limonene and its
products. Thus, we believe this leads to the different results regarding the formation of HOMs and dimers.
Regarding $\Delta^3$-carene, the mechanisms and processes are almost opposite to those of limonene. First,
water vapor reacts with $sCI_1$ or $sCI_2$ to promote the formation of α-hydroxyalkyl-hydroperoxides (Fig.
S5). Their subsequent products without second ozonolysis of exocyclic double bond have higher
volatility, and may most likely prevail in the gas phase. In addition, it has been found that α-hydroxyalkyl
hydroperoxides preferentially decompose into aldehydes and $H_2O_2$ (Kumar et al., 2014; Chen et al., 2016),
i.e., 3-caronaldehyde for $\Delta^3$-carene, which has higher volatility than the products from other reaction
pathways. Correspondingly, the number and relative intensity of HOMs and dimers detected under high
RH conditions are both lower than those under low RH conditions (Table S8). Furthermore, out of a total
of 178 dimers, 63 dimers were exclusively identified under low RH conditions (Table S7). As a result,
high RH shows an inhibitory effect on the SOA formation from $\Delta^3$-carene ozonolysis.
To investigate the multi-generation reactions of limonene under low-concentration conditions, we
conducted low-concentration limonene ozonolysis experiments, and the results are shown in Fig. S6. In
these experiments, the limonene and $O_3$ concentrations were 20.5 ppb and 5.7 ppm, respectively.
According to the experimental results, the number concentration of SOA formed from limonene
ozonolysis increased by approximately 1.4 times under high RH, which is similar to the increase observed
under high-loading conditions. The mass concentration increased by approximately 1.3 times at a
precursor concentration of 20.5 ppb. The relatively small increase in mass concentration compared to the
high-concentration conditions may be attributed to the less pronounced distribution of SVOCs at low
mass concentrations. This result suggests that the enhancement effect on limonene SOA by high RH is
still valid for low precursor concentrations.
To further confirm the assumption that water-influenced multi-generation reactions of the exocyclic
double bond enhance the SOA formation, we conducted two comparative analyses: firstly, we examined
the ozonolysis of the endocyclic double bond in limonene, leaving the exocyclic double bond unreacted.
This was done by applying a low $O_3$ concentration (~67 ppb), since the reaction of $O_3$ with endocyclic
double bond is ~30 times faster than the reaction of $O_3$ with exocyclic double bond (Shu and Atkinson,
1994). Interestingly, when limonene was oxidized at only the endocyclic double bond, we observed a
slight decrease in both the number and mass concentrations as the RH increased (Fig. S7). This is similar
to the results obtained for $\Delta^3$-carene, which contains only one endocyclic double bond. Secondly, we
compared the ozonolysis of structurally similar β-caryophyllene, which has an exocyclic C-C double
bond that can undergo further reactions (Fig. S8). As expected, we observe a large enhancement in SOA
formation under high RH condition (Table S9 and Fig. S9). This implies that monoterpenes,
sesquiterpenes, and other BVOCs with two unsaturation double bonds may follow similar reaction
mechanisms during ozonolysis, and thus have a RH dependency in SOA production.
**4 Conclusions**
In this study, the effect of humidity on SOA production from the ozonolysis of two monoterpenes
(limonene and $\Delta^3$-carene) was investigated with an OFR. Contrasting impacts of RH on the SOA
formation were observed: limonene-SOA yield increases by ~100% when RH changes from ~1% to
~60%, while $\Delta^3$-carene-SOA yield slightly decreases. By analyzing the chemical composition of SOA
with ESI-Q-TOF-MS, we find that the multi-generation reactions of the exocyclic C-C double bond are
likely the driving force of the enhancement in limonene-SOA. The presence of water promotes the
formation of carbonyls from the reaction of exocyclic double bond, and further favors the formation of
dimers and HOMs. This leads to promoted new particle formation and subsequent condensation of
SVOCs. These reactions also lower the volatilities of the SVOCs, and further promote the gas-particle
partitioning. Moreover, this hypothesis is supported by a similar behavior of the ozonolysis of β-
caryophyllene (sesquiterpene with an exocyclic double bond) in SOA enhancement under high RH
condition. However, since aerosol dynamics of small clusters and particles are very complex, we do not
rule out a series of reactions that may occur in the particle phase. The results in this study suggest that
multi-generation reactions play an important role in SOA formation from the ozonolysis of BVOCs,
which are significantly influenced by humidity. This impact is largely dependent on the molecular
structure of the SOA precursors (e.g., with or without the exocyclic double bond), thus highlighting the
importance to consider the molecular structure of monoterpenes in modeling and field studies of biogenic
SOA.

**Data availability.** Experimental data are available upon request to the corresponding authors.
**Supplement.** The supplement related to this article is available online.
**Author contributions.** LD and SZ designed the experiments and SZ carried them out. SZ performed
data analysis with assistance from KL, LD, ZY, and JL. SZ and KL wrote the paper with contributions
from all co-authors.
**Declaration**. The authors declare that they have no conflict of interest.
**Acknowledgements.** We thank Guannan Lin, Jingyao Qu and Zhifeng Li from the State Key Laboratory
of Microbial Technology of Shandong University for help and guidance with MS measurements.
**Financial support.** This research has been supported by the National Natural Science Foundation of
China (grant no. 22076099), and the Fundamental Research Fund of Shandong University (grant no.
2020QNQT012).

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
