# Peer review of "Contrasting impacts of humidity on the ozonolysis of monoterpenes: insights into the multi-generation chemical mechanism"

_EGUsphere, 2023_

## Author Comment (AC1)

**Response to the comments of Anonymous Referee #1**

General comments:

Zhang et al used an oxidation flow reactor (OFR) to produce SOA at different relative humidities, and attempt to explain the changes in SOA loadings caused by the changing RH. Unfortunately, I do not find their explanations and conclusions to be based on solid evidence or argumentation. I also cannot envision that further analysis of their data would allow new insights into the topic of how humidity influences SOA formation. Therefore, I cannot suggest this manuscript for publication in ACP. I outline some of the main shortcomings below.

Response: We thank the Referee for their feedback on our manuscript. We have carefully considered the comments and have made significant revisions to address the concerns that the Referee raised.

In our revised version, we have conducted exocyclic double bond limonene ozonolysis experiments to verify the RH effects on multi-generation reactions of limonene. We have also added low concentration limonene experiments to provide further insights at low SOA loadings. In addition, we have revised the proposed mechanism and analysis for limonene and $\Delta^3$-carene ozonolysis. The responses are listed below in blue color text and the associated revisions to the manuscript are shown in red color text.

Major comments

1. The experiments were conducted at loadings far from atmospheric concentrations, yet this aspect, and how it might impact the relevance of the study to the atmosphere, was not discussed at all. The gas phase oxidation chemistry of monoterpenes is a complicated process, largely due to competing fates of $RO_2$ radicals under different conditions, and further reactions are very likely to take place after condensation into SOA (e.g. Pospisilova et al., 2020; Kalberer et al., 2004). As such, using offline SOA composition data to infer something about the reactions taking place in the first milliseconds after oxidant attack is extremely challenging and would, at the very least, require detailed analyses to exclude that any of the other stages of potential reactions are negligible. This was not done, and I cannot see that it would be possible with the data available in this study. In fact, the offline MS data is even said to stay relatively unchanged with the change in RH (line 194), so the chemical insights will be very limited, and really the only data used to draw conclusions on is the SMPS data. This is simply not enough for reaching any conclusive chemical understanding of the processes.

Response: To get a better understanding of the multi-generation reactions of limonene under low loadings, we have conducted a low-concentration limonene ozonolysis experiment (Fig. S6). The results of this experiment also revealed an enhancement effect of RH on limonene SOA similar to that observed at high loadings. This has been updated in the revised manuscript at Page 18, Line 339-348: "To investigate the multigeneration reactions of limonene under low-concentration conditions, we conducted low-concentration limonene ozonolysis experiments, and the results are shown in Fig. S6. In these experiments, the limonene and $O_3$ concentrations were 20.5 ppb and 5.7 ppm, respectively. According to the experimental results, the number concentration of SOA formed from limonene ozonolysis increased by approximately 1.4 times under high RH, which is similar to the increase observed under high-loading conditions. The mass concentration increased by approximately 1.3 times at a precursor concentration of 20.5 ppb. The relatively small increase in mass concentration compared to the high-concentration conditions may be attributed to the less pronounced distribution of SVOCs at low mass concentrations. This result indicates that the enhancement effect on limonene SOA by high RH is still valid for low precursor concentrations."

[Figure]

**Figure S6.** The SOA formation of low-concentration limonene under low and high RH (a) mass concentration (b) number concentration (c) SOA yield (d) mean diameter.

Offline ESI mass spectrometry analysis of particulate matter is an effective technique that provides valuable information about the SOA formed from the oxidation of VOCs. This technique is powerful to provide the molecular composition of SOA and enable the determination of the formation mechanisms of the oxidation products, including gas-phase reaction products. Indeed, many previous studies have applied off-line ESI-MS for the identification of SOA components and their formation mechanism. For instance, Zhao et al. (2022) proposed the gas-phase formation pathways of dimer esters in SOA arising from the ozonolyisis of α-pinene using the offline analysis with UPLC-QTOF-MS. Furthermore, employing offline HPLC/ESI-TOF-MS analysis, Iinuma et al. (2007) proposed a detailed formation mechanism for two organosulfates associated with SOA formation. Additionally, Li et al. (2020) proposed gas-phase OH oxidation mechanism of long-chain alkanes using offline ESI-TOF-MS. Similarly, through offline UPLC/ESI-TOF-MS analysis, Thomsen et al. (2021) inferred that the α-pinene-derived analogue, cis-pinic acid, tends to stay in the gas phase and undergo further reactions before condensing.

In order to minimize the potential impact of post-collection reactions on our

experimental results, we employed a consistent treatment across all experiments, where the collected particles are immediately dissolved to avoid any potential influence of particle-phase reactions under different experimental conditions.

We did not observe significant changes in the type of mass spectrum peaks between low and high RH. However, we did observe variations in the peak intensities of certain specific products, which can be attributed to the influence of RH. These effects promote specific reaction pathways, thereby facilitating the formation of corresponding products.

In addition, to further verify the hypothesis regarding the influence of water on multi-generation reactions of exocyclic double bonds, we have conducted the endo-double bonds limonene ozonolysis under low $O_3$ concentration (67 ppb) and high precursor concertation (450 ppb). Under this condition, the ozonolysis mostly happened for the endocyclic double bond in limonene, leaving the exocyclic double bond almost unreacted, since the reaction of $O_3$ with endocyclic double bond is ~30 times faster than the reaction of $O_3$ with exocyclic double bond (Shu and Atkinson, 1994). As expected, when limonene was oxidized at only the endocyclic double bond, we observed a slight decrease in both the number and mass concentrations as the RH increased (Fig. S7). This is similar to the results obtained for $\Delta^3$-carene, which contains only an endocyclic double bond. The results of these control experiments provide more evidence that the multi-generation reactions play important roles in the limonene SOA enhancement by high RH. This was revised it in Page18, Line 349-356: "To further confirm the assumption that water-influenced multi-generation reactions of the exocyclic double bond enhance the SOA formation, we conducted two comparative analyses: firstly, we examined the ozonolysis of the endocyclic double bond in limonene, leaving the exocyclic double bond unreacted. This was done by applying a low $O_3$ concentration (~67 ppb), since the reaction of $O_3$ with endocyclic double bond is ~30 times faster than the reaction of $O_3$ with exocyclic double bond (Shu and Atkinson, 1994). Interestingly, when limonene was oxidized at only the endocyclic double bond, we observed a slight decrease in both the number and mass concentrations as the RH increased (Fig. S7). This is similar to the results obtained for $\Delta^3$-carene, which contains only one endocyclic double bond."

[Figure]

**Figure S7.** The SOA formation from endocyclic ozonolysis of limonene under low and high RH (a) mass concentration (b) number concentration (c) SOA yield (d) mean diameter. The initial concentration of limonene is 450 ppb and the concentration of $O_3$ is 67 ppb. Limonene ozonolysis primarily took place on endo-double bonds, with a rate constant of $2.01 \times 10^{-16}$ $cm^3$ $molec^{-1}$ $s^{-1}$ (Shu and Atkinson, 1994). Based on this rate constant, it can be estimated that approximately 10% of the limonene was consumed by $O_3$ upon exiting the reactor.

In summary, by keeping the same protocols in sample processing and analysis, the errors introduced from off-line analysis were minimized. The differences in specific products in the MS indicate the RH effects on the subsequent reactions after the ozonolysis of the exocyclic double bond. This mechanism was further verified by the control experiments at low ozone concentration. Therefore, we believe that the results here can improve our understanding of the multi-generation chemical processes of limonene.

2. There are several studies concluding that RH does not have a noticeable impact on the formation of the most oxidized products that are expected to contribute most to SOA (e.g. Surdu et al., 2023; Li et al., 2019). Surdu et al. also analyze the potential reasons for the RH-driven changes in relation to particle phase reactions and changes in partitioning.

Response: We acknowledge that relative humidity (RH) was reported not to have a noticeable impact on the formation of the most oxidized products such as in Surdu et al. (2023). However, it is important to note that this study was conducted under photooxidation conditions whereas our experiment was conducted under dark conditions. Furthermore, their investigation of SOA formation was carried out to achieve a steady-state particle growth, with RH gradually increasing while maintaining all other experimental conditions constant. In contrast, our experiment focused on exploring the influence of water throughout the entire oxidation process of SOA by using different initial degrees of humidity.

With regard to Li et al. (2019), although both their study and ours investigate ozonolysis, the experimental conditions are quite different. The ozone concentration and residence time of the reactor are 900 ppb and 60 s in Li et al. (2019), while they are ~6 ppm and 167 s in this study. This leads to a ~18 times higher ozone exposure in this study. In addition, the limonene concentrations are 1085 ppb and 321 ppb in Li et al, 2019 and in this study, respectively. All these differences lead to an easier ozone reaction of the exocyclic double bond in our study. In addition, the high ozone/limonene ratio in this study is believed to be more similar to the real atmospheric conditions. We have revised this in the revised manuscript in Page 17, Line 322-328: "In contrast, Li et al. (2019) found negligible change in dimers and HOMs in limonene-$O_3$ system when changing RH from 0 to 60%. The discrepancy is mainly attributed to the different experimental conditions. The ozone exposure in this study is ~18 times higher than in Li et al. (2019), while the limonene concentration in this study is only ~30% of that in their study. These two conditions both favor the multi-generation reactions occurred at

the exocyclic double bond of limonene and its products. Thus, we believe this leads to the different results regarding the formation of HOMs and dimers."

3. In addition to their speculative nature, the chemical mechanisms drawn up and discussed are wrong/misleading concerning the sCI. Only a small part of the formed CI will stabilize (and thus be impacted by RH), as most of them will simply decompose through the typical vinyl hydroperoxide channel. Reaction with water vapor is normally only relevant for the stabilized CI. In this manuscript, it is proposed that ozonolysis produces sCI at a 100% yield (e.g. lines 206-208, Fig 3-4). This raises further questions concerning how well the authors have understood the reactions that they are using to explain their observations.

Response: We have redrawn Figures 3, 4 and S5 and revised the corresponding mechanisms. In the revised figures, the POZ formed from the reaction of the endocyclic double bond generates $eCI_1$ and $eCI_2$ with branching ratios of 0.35 and 0.65, respectively. The reaction pathways associated with eCIs are complex. However, the generation of sCIs and the OH pathway are believed to be dominant (Nguyen et al., 2016). It is inferred that about 46% of eCIs formed from $\alpha$-pinene ozonolysis would stabilize (Tillmann et al., 2010). Furthermore, it has been observed that the presence of OH plays a significant role in promoting the formation of HOMs (Crounse et al., 2013). Therefore, in this study, we focus our discussion on the sCI and OH pathways which are illustrated in the proposed mechanism depicted in Figures 3 and 4.

These details have been updated in the revised manuscript at Page 12 as follows:

Line 218-221:
"In the context of $eCI_1$, several complex reactions occur, with the most dominant reaction being the generation of hydroxyl radicals (OH) and a reaction pathway known as $sCI_1$. The $sCI_1$ pathway can proceed through three distinct reactions, as depicted in Fig. 3."

Line 224-226:
"The second and third pathways involve reactions of $sCI_1$ with carboxylic acids and carbonyls, respectively, leading to the formation of anhydrides and secondary ozonides. Additionally, the generated OH radicals can react with limonene, giving rise to another alkyl radical, $C_{10}H_{17}O\cdot$."

Line 236-238:
"In addition to the $eCI_1$ route, the $eCI_2$ pathway is also responsible for the generation of various products (Fig. 4). Since the reaction of the hydroxyl radical (OH) attacking limonene is already depicted in Fig. 3, our main emphasis in Fig. 4 is on the pathways involved in the generation of SCI."

[Figure]

**Figure 3.** Proposed formation mechanism for SOA formation from eCI$_1$ oxidation under high RH. The compounds in blue and in boxes are identified using UPLC/ (−) ESI-Q-TOF-MS.

**Figure 4.** Proposed formation mechanisms for SOA formation from eCI$_2$ and exocyclic double bond oxidation under high RH. The compounds in blue and in boxes are identified using UPLC/ (−) ESI-Q-TOF-MS.

[Figure]

**Figure S5.** Proposed formation mechanisms for SOA formation from $\Delta^3$-carene ozonolysis under high RH.

4. In addition to the major concerns above, there are various other question marks concerning the conclusions drawn. The reasoning is not very clear concerning how the effect of increased sCI+H$_2$O reactions would cause the observed changes. The main argumentation seems to be that the yield of carbonyls increases, which also makes the oligomerization more efficient. However, for conclusions like this, there should be more clearly stated what the proposed reactions are and, even more importantly, what are the competing reaction pathways (which then should produce something else, with a lower SOA yield).

Response: In this study, limonene, due to its extra exocyclic double bond, undergoes water-induced oxidation to form carbonyls. This oxidation process lowers its volatility, which results in a higher overall mass concentration compared to $\Delta^3$-carene. Specially, the oligomerization of these carbonyls generates more dimers including hemiacetal (or acetal) formation and aldol condensation (Zhang et al., 2022; Kroll et al., 2005; Jang et al., 2003). Correspondingly, it was found that 54 out of the total 187 dimers were exclusively observed under high humidity conditions for limonene-SOA (Table S6). We have added the following content in the revised manuscript (Page 15, Line 273-277): "and the oligomerization of these carbonyls generates more dimers including hemiacetal (or acetal) formation and aldol condensation (Zhang et al., 2022; Kroll et al., 2005; Jang et al., 2003). As shown in Table S6, 54 out of the total 187 dimers were exclusively observed for limonene under high humidity conditions, contributing to a corresponding intensity of ~19%."

**Table S6.** Dimers: RH-dependent discoveries for limonene and $\Delta^3$-carene.

| 54 dimers exclusively detected under high RH (limonene) | | 63 dimers exclusively detected under low RH ($\Delta^3$-carene) | |
|---|---|---|---|
| Molecular formula | Absolute intensity (High RH) | Molecular formula | Absolute intensity (Low RH) |
| $C_{18}H_{26}O_4$ | $4.66 \times 10^2$ | $C_{17}H_{24}O_5$ | $1.59 \times 10^3$ |

| Formula | Value | Formula | Value |
|---|---|---|---|
| $C_{16}H_{20}O_6$ | $7.24\times10^2$ | $C_{10}H_{14}O_{11}$ | $3.90\times10^3$ |
| $C_{13}H_{18}O_9$ | $3.36\times10^2$ | $C_{14}H_{14}O_8$ | $4.02\times10^3$ |
| $C_{17}H_{22}O_6$ | $6.63\times10^3$ | $C_{20}H_{40}O_2$ | $4.60\times10^3$ |
| $C_{18}H_{26}O_5$ | $6.28\times10^2$ | $C_{12}H_{10}O_{10}$ | $4.00\times10^3$ |
| $C_{19}H_{32}O_4$ | $1.58\times10^3$ | $C_{13}H_{16}O_9$ | $8.34\times10^3$ |
| $C_{15}H_{18}O_8$ | $1.65\times10^3$ | $C_{19}H_{26}O_4$ | $4.96\times10^3$ |
| $C_{13}H_{12}O_{10}$ | $8.85\times10^3$ | $C_{17}H_{22}O_6$ | $1.05\times10^3$ |
| $C_{14}H_{20}O_9$ | $8.44\times10^2$ | $C_{13}H_{12}O_{10}$ | $5.46\times10^3$ |
| $C_{16}H_{28}O_7$ | $9.89\times10^3$ | $C_{13}H_{18}O_{10}$ | $4.68\times10^3$ |
| $C_{15}H_{26}O_8$ | $2.18\times10^3$ | $C_{15}H_{12}O_9$ | $4.22\times10^3$ |
| $C_{10}H_8O_{13}$ | $6.33\times10^3$ | $C_{10}H_{12}O_{13}$ | $5.00\times10^3$ |
| $C_{18}H_{24}O_6$ | $6.06\times10^2$ | $C_{22}H_{28}O_3$ | $8.88\times10^3$ |
| $C_{11}H_{14}O_{12}$ | $7.70\times10^2$ | $C_{19}H_{26}O_6$ | $1.54\times10^3$ |
| $C_{21}H_{22}O_4$ | $4.80\times10^3$ | $C_{16}H_{20}O_9$ | $1.64\times10^3$ |
| C20H34O4 | $2.53\times10^3$ | $C_{15}H_{18}O_{10}$ | $5.00\times10^3$ |
| $C_{23}H_{32}O_2$ | $2.12\times10^3$ | $C_{16}H_{22}O_9$ | $1.69\times10^3$ |
| $C_{18}H_{32}O_6$ | $3.68\times10^2$ | $C_{18}H_{22}O_8$ | $3.32\times10^3$ |
| $C_{17}H_{30}O_7$ | $7.46\times10^3$ | $C_{12}H_{16}O_{13}$ | $4.00\times10^3$ |
| $C_{14}H_{22}O_{10}$ | $4.04\times10^3$ | $C_{20}H_{32}O_6$ | $8.21\times10^3$ |
| $C_{21}H_{36}O_4$ | $1.36\times10^4$ | $C_{16}H_{18}O_{10}$ | $4.50\times10^3$ |
| $C_{17}H_{30}O_8$ | $4.68\times10^2$ | $C_{16}H_{20}O_{10}$ | $5.20\times10^3$ |
| $C_{12}H_{16}O_{13}$ | $2.43\times10^3$ | $C_{19}H_{24}O_8$ | $8.21\times10^3$ |
| $C_{11}H_{14}O_{14}$ | $4.46\times10^2$ | $C_{20}H_{28}O_7$ | $2.38\times10^3$ |
| $C_{18}H_{30}O_8$ | $4.46\times10^2$ | $C_{17}H_{20}O_{10}$ | $4.16\times10^3$ |
| $C_{16}H_{26}O_{10}$ | $7.44\times10^2$ | $C_{21}H_{36}O_6$ | $8.03\times10^3$ |
| $C_{17}H_{20}O_{10}$ | $2.12\times10^3$ | $C_{16}H_{26}O_{11}$ | $1.16\times10^3$ |
| $C_{16}H_{24}O_{11}$ | $1.48\times10^3$ | $C_{17}H_{26}O_{11}$ | $1.32\times10^3$ |
| $C_{20}H_{24}O_8$ | $3.96\times10^3$ | $C_{18}H_{18}O_{11}$ | $4.02\times10^3$ |
| $C_{17}H_{22}O_{11}$ | $2.48\times10^3$ | $C_{18}H_{22}O_{11}$ | $4.54\times10^3$ |
| $C_{21}H_{34}O_8$ | $1.28\times10^4$ | $C_{18}H_{26}O_{11}$ | $1.49\times10^3$ |
| $C_{13}H_{22}O_{15}$ | $4.06\times10^2$ | $C_{22}H_{28}O_8$ | $4.62\times10^3$ |
| $C_{19}H_{32}O_{10}$ | $5.30\times10^2$ | $C_{15}H_{18}O_{14}$ | $4.08\times10^3$ |
| $C_{22}H_{32}O_8$ | $5.90\times10^3$ | $C_{20}H_{32}O_{10}$ | $5.97\times10^3$ |
| $C_{20}H_{28}O_{10}$ | $1.53\times10^3$ | $C_{17}H_{22}O_{13}$ | $5.10\times10^3$ |
| $C_{18}H_{18}O_{13}$ | $4.49\times10^3$ | $C_{21}H_{28}O_{10}$ | $4.25\times10^3$ |
| $C_{19}H_{24}O_{12}$ | $1.49\times10^4$ | $C_{19}H_{22}O_{12}$ | $5.44\times10^3$ |
| $C_{19}H_{30}O_{12}$ | $6.10\times10^2$ | $C_{22}H_{34}O_9$ | $7.52\times10^3$ |
| $C_{15}H_{18}O_{16}$ | $1.14\times10^3$ | $C_{21}H_{34}O_{10}$ | $2.12\times10^3$ |
| $C_{23}H_{38}O_9$ | $4.34\times10^2$ | $C_{14}H_{24}O_{16}$ | $4.80\times10^3$ |
| $C_{32}H_{44}O_2$ | $8.96\times10^2$ | $C_{15}H_{22}O_{16}$ | $4.04\times10^3$ |
| $C_{21}H_{36}O_{11}$ | $3.74\times10^2$ | $C_{17}H_{30}O_{14}$ | $3.51\times10^3$ |
| $C_{14}H_{26}O_{17}$ | $1.00\times10^3$ | $C_{22}H_{36}O_{10}$ | $4.02\times10^3$ |
| $C_{20}H_{26}O_{13}$ | $1.26\times10^4$ | $C_{18}H_{24}O_{14}$ | $4.44\times10^3$ |

| | | | |
|---|---|---|---|
| $C_{22}H_{34}O_{11}$ | $1.92\times10^3$ | $C_{19}H_{28}O_{13}$ | $6.68\times10^3$ |
| $C_{20}H_{30}O_{13}$ | $9.36\times10^2$ | $C_{20}H_{22}O_{13}$ | $3.90\times10^3$ |
| $C_{18}H_{24}O_{15}$ | $2.05\times10^3$ | $C_{21}H_{26}O_{12}$ | $4.48\times10^3$ |
| $C_{21}H_{38}O_{12}$ | $9.16\times10^2$ | $C_{22}H_{30}O_{11}$ | $2.29\times10^3$ |
| $C_{24}H_{38}O_{10}$ | $3.78\times10^3$ | $C_{15}H_{24}O_{17}$ | $4.70\times10^3$ |
| $C_{16}H_{24}O_{17}$ | $1.26\times10^3$ | $C_{25}H_{38}O_9$ | $5.24\times10^3$ |
| $C_{21}H_{24}O_{14}$ | $4.80\times10^3$ | $C_{17}H_{26}O_{16}$ | $5.18\times10^3$ |
| $C_{20}H_{34}O_4$ | $4.98\times10^2$ | $C_{21}H_{26}O_{13}$ | $4.82\times10^3$ |
| $C_{18}H_{30}O_6$ | $2.74\times10^3$ | $C_{22}H_{30}O_{12}$ | $2.47\times10^3$ |
| $C_{18}H_{28}O_7$ | $1.53\times10^4$ | $C_{16}H_{24}O_{17}$ | $5.16\times10^3$ |
| | | $C_{17}H_{28}O_{16}$ | $6.58\times10^3$ |
| | | $C_{29}H_{44}O_6$ | $5.82\times10^3$ |
| | | $C_{17}H_{30}O_{16}$ | $2.06\times10^3$ |
| | | $C_{22}H_{38}O_{12}$ | $3.86\times10^3$ |
| | | $C_{16}H_{32}O_{17}$ | $7.04\times10^3$ |
| | | $C_{23}H_{30}O_{12}$ | $1.26\times10^3$ |
| | | $C_{24}H_{34}O_{11}$ | $6.82\times10^3$ |
| | | $C_{20}H_{30}O_{10}$ | $4.14\times10^3$ |
| | | $C_{20}H_{32}O_{11}$ | $3.41\times10^3$ |

**Reference**

Crounse, J. D., Nielsen, L. B., Jørgensen, S., Kjaergaard, H. G., and Wennberg, P. O.: Autoxidation of Organic Compounds in the Atmosphere, J. Phys. Chem. Lett., 4, 3513-3520, 10.1021/jz4019207, 2013.

Iinuma, Y., Müller, C., Berndt, T., Böge, O., Claeys, M., and Herrmann, H.: Evidence for the Existence of Organosulfates from β-Pinene Ozonolysis in Ambient Secondary Organic Aerosol, Environ. Sci. Technol., 41, 6678-6683, 10.1021/es070938t, 2007.

Jang, M. S., Carroll, B., Chandramouli, B., and Kamens, R. M.: Particle growth by acid-catalyzed heterogeneous reactions of organic carbonyls on preexisting aerosols, Environ. Sci. Technol., 37, 3828-3837, 10.1021/es021005u, 2003.

Kroll, J. H., Ng, N. L., Murphy, S. M., Varutbangkul, V., Flagan, R. C., and Seinfeld, J. H.: Chamber studies of secondary organic aerosol growth by reactive uptake of simple carbonyl compounds, J. Geophys. Res.-Atmos., 110, 10.1029/2005JD006004, 2005.

Li, J., Wang, W., Li, K., Zhang, W., Peng, C., Zhou, L., Shi, B., Chen, Y., Liu, M., Li, H., and Ge, M.: Temperature effects on optical properties and chemical composition of secondary organic aerosol derived from n-dodecane, Atmos. Chem. Phys., 20, 8123-8137, 10.5194/acp-20-8123-2020, 2020.

Li, X., Chee, S., Hao, J., Abbatt, J. P. D., Jiang, J., and Smith, J. N.: Relative humidity effect on the formation of highly oxidized molecules and new particles during monoterpene oxidation, Atmos. Chem. Phys., 19, 1555-1570, https://doi.org/10.5194/acp-19-1555-2019, 2019.

Nguyen, T. B., Tyndall, G. S., Crounse, J. D., Teng, A. P., Bates, K. H., Schwantes, R. H., Coggon, M. M., Zhang, L., Feiner, P., Milller, D. O., Skog, K. M., Rivera-Rios, J. C., Dorris, M., Olson, K. F., Koss, A., Wild, R. J., Brown, S. S., Goldstein, A. H., de Gouw, J. A., Brune, W. H., Keutsch, F. N., Seinfeldcj, J. H., and Wennberg, P. O.: Atmospheric fates of Criegee intermediates in the ozonolysis of isoprene, Phys. Chem. Chem. Phys., 18, 10241-10254, 10.1039/c6cp00053c, 2016.

Shu, Y. G. and Atkinson, R.: RATE CONSTANTS FOR THE GAS-PHASE REACTIONS OF O-3 WITH A SERIES OF TERPENES AND OH RADICAL FORMATION FROM THE O-3 REACTIONS WITH SESQUITERPENES AT 296+/-2-K, INTERNATIONAL JOURNAL OF CHEMICAL KINETICS, 26, 1193-1205, 10.1002/kin.550261207, 1994.

Surdu, M., Lamkaddam, H., Wang, D. S., Bell, D. M., Xiao, M., Lee, C. P., Li, D., Caudillo, L., Marie, G., Scholz, W., Wang, M., Lopez, B., Piedehierro, A. A., Ataei, F., Baalbaki, R., Bertozzi, B., Bogert, P., Brasseur, Z., Dada, L., Duplissy, J., Finkenzeller, H., He, X.-C., Höhler, K., Korhonen, K., Krechmer, J. E., Lehtipalo, K., Mahfouz, N. G. A., Manninen, H. E., Marten, R., Massabò, D., Mauldin, R., Petäjä, T., Pfeifer, J., Philippov, M., Rörup, B., Simon, M., Shen, J., Umo, N. S., Vogel, F., Weber, S. K., Zauner-Wieczorek, M., Volkamer, R., Saathoff, H., Möhler, O., Kirkby, J., Worsnop, D. R., Kulmala, M., Stratmann, F., Hansel, A., Curtius, J., Welti, A., Riva, M., Donahue, N. M., Baltensperger, U., and El Haddad, I.:

Molecular Understanding of the Enhancement in Organic Aerosol Mass at High Relative Humidity, Environ. Sci. Technol., 57, 2297-2309, 10.1021/acs.est.2c04587, 2023.

Thomsen, D., Elm, J., Rosati, B., Skønager, J. T., Bilde, M., and Glasius, M.: Large Discrepancy in the Formation of Secondary Organic Aerosols from Structurally Similar Monoterpenes, ACS Earth Space Chem., 5, 632-644, 10.1021/acsearthspacechem.0c00332, 2021.

Tillmann, R., Hallquist, M., Jonsson, Å. M., Kiendler-Scharr, A., Saathoff, H., Iinuma, Y., and Mentel, T. F.: Influence of relative humidity and temperature on the production of pinonaldehyde and OH radicals from the ozonolysis of α-pinene, Atmos. Chem. Phys., 10, 7057-7072, 10.5194/acp-10-7057-2010, 2010.

Zhang, Y., He, L., Sun, X., Ventura, O. N., and Herrmann, H.: Theoretical Investigation on the Oligomerization of Methylglyoxal and Glyoxal in Aqueous Atmospheric Aerosol Particles, ACS Earth Space Chem., 6, 1031-1043, 10.1021/acsearthspacechem.1c00422, 2022.

Zhao, Y., Yao, M., Wang, Y., Li, Z., Wang, S., Li, C., and Xiao, H.: Acylperoxy Radicals as Key Intermediates in the Formation of Dimeric Compounds in α-Pinene Secondary Organic Aerosol, Environ. Sci. Technol., 56, 14249-14261, 10.1021/acs.est.2c02090, 2022.

---

## Author Comment (AC2)

**Response to the comments of Anonymous Referee #2**

General comments:

In the study by Zhang et al., the authors explore the effect of humidity (RH) on the formation of secondary organic aerosol (SOA) from ozonolysis of two structurally different monoterpenes; limonene and $\Delta^3$-carene, and the sesquiterpene β-caryophyllene. Experiments are performed at constant temperature in an oxidation flow reactor at RHs ranging from 1-60 % whilst monitoring SOA particle number and mass concentrations followed by off-line analyses of the SOA chemical composition using ultra-high performance liquid chromatography quadrupole time-of-flight mass spectrometry (LC-MS). The study reports large differences in the effect of RH on the SOA formation from limonene and $\Delta^3$-carene, with the former showing increasing SOA mass and particle number concentration at elevated RH whilst little or no effects are observed in the case of $\Delta^3$-carene. From the chemical composition of the formed SOA the authors explain these discrepancies by water-influenced reactions on exocyclic double bonds yielding lower volatile organic compounds under higher RH.

The topic is relevant and fall within the scope of ACP, however, the manuscript is in need of major revision before any consideration for publication in ACP

Major concerns include the far from atmospheric relevant conditions applied in the study experiments, lack of validation of experimental approach, lack of discussion on contribution of other oxidizing agents, as well as scarce evidence of enhanced dimer formation at elevated RH from chemical analysis of formed the SOA. These are concerns that needs to be addressed if the manuscript is in any way to contribute to the work of the many previous studies reporting on the influence of RH on the formation of SOA from monoterpenes.

Response: We thank the Referee for providing the feedback on our manuscript. We have carefully considered all of the concerns raised by the Referee and made revisions accordingly. Below are the specific changes we have made:

1. We have conducted a low-concentration limonene ozonolysis experiment to better simulate atmospheric conditions and enhance the relevance of our findings.

2. In order to give an evidence of enhanced dimer formation at elevated RH from chemical analysis of formed the SOA, we have reanalyzed the distribution of the dimers formed by limonene and $\Delta^3$-carene ozonolysis. Additionally, we have added the distribution of the monomers, dimers, trimer and tetramer in the overall mass spectrometry spectrum of $\Delta^3$-carene.

The responses are listed below in blue color text and the associated revisions to the manuscript are shown in red color text.

Major comments:

1. As I understand, this is the first publication using the custom-made oxidation flow

reactor (OFR). With a length of 6.02 meters, have the authors validated that measurements of e.g. ozone, RH, temperature and VOC performed at the end of the OFR represent initial conditions at the point of injection and thus initial oxidation? Other OFRs, such as in Jonsson et al., (2006) and Li et al., (2019), is designed to ensure proper mixing of injected oxidant (e.g. $O_3$) and VOCs at the initial stage of the OFR. When using OFRs the uniform distributions of $O_3$, VOCs and $H_2O$ in the tube should be confirmed by measuring $O_3$, VOC and RH at the different locations prior to the experiments. A particular concern is that the $O_3$: VOC ratio and maybe RH may be different at the point of injection compared to the end of the 6.02 meter tube.

Response: We are sorry for the oversight in labeling the dimensions of the oxidation flow reactor. It should have been specified as 602 mm, and we have made the correction in the revised version of the manuscript, as indicated by the highlighted yellow text (Page 3, Line 80). This OFR is similar to the design shown in previous studies (Liu et al., 2019; Liu et al., 2014), containing a mixing tank and a reacting tube, which has been proved to have good mixing ability. We have added more descriptions of OFR (Fig. S2) and related references in the revised manuscript.

"The OFR is a 602 mm long stainless cylinder with a volume of 2.5 L (Fig. S2) (Liu et al., 2014; Liu et al., 2019)"

[Figure]

**Figure S2.** Schematic description of the experiment.

2. The authors report SOA mass concentrations of 980-2200 ug/m$^3$ from the oxidation of 321 ppb of limonene by 6 ppm of $O_3$ with corresponding yields of 63-142% (table 1). These values are very high in comparison with other studies which should be made apparent by the authors. E.g. for clarification, please add mass concentrations and yields of all studies in table 2. Any explanation for these high yields?

Response: We have added $O_3$ concentration, SOA mass concentration and yield in Table 2. As a comparison, the ozonolysis of 13.2 ppb of limonene (Δorg) with 430 ppb of $O_3$ resulted in concentration ranges of 62-229 μg/m$^3$, and the corresponding yields ranged from 77.4% to 285.7% (Jonsson et al., 2006). In addition, the SOA potential of the exocyclic bond was found to be relatively high. Specifically, the SOA yield from the exocyclic bond was up to eight times higher compared to the endocyclic bond, with corresponding yields of approximately 23.8%-55.3% and 7.4%, respectively (Gong and Chen, 2021). Due to high limonene and $O_3$ concentrations, the SOA yields in this study are relatively high, but still in the range that was previously reported.

**Table 2.** Comparison with previous studies on the effect of RH.

| Precursor | Precursor concentration (ppb) | O₃ concentration (ppb) | Reactor | OH scavenger | T (K) | RH (%) | SOA Mass Concentration (µg/m³) | SOA Yield (%) | Mᵃ | Nᵇ | Reference |
|---|---|---|---|---|---|---|---|---|---|---|---|
| | 1000 | 1000 | flow reactor | cyclohexane | 295±2 | 0.02 and 32.5 | N.Mᶜ | N.Mᶜ | no effect | –ᵉ | Bonn et al. (2002) |
| | 320 | 100±5 | chamber | N.Mᶜ | 296±2 | 18±2, 50±3 and 82±2 | 24; 58; 120 | 7.0±0.7; 17.4±1.3; 53.4±1.9 | +ᵈ (7 times) | +ᵈ (8 times) | Yu et al. (2011) |
| limonene | 15 and 30 | 430.9 | flow reactor | 2-butanol | 298±0.4 | <2-85 | 2.7-10.5 and 62-229 | 6.8-26.4 and 77.4-285.7 | +ᵈ | +ᵈ | Jonsson et al. (2006) |
| | endocyclic (24.6) and exocyclic | endocyclic (270) and exocyclic | flow reactor | 2-butanol | 298 | 10-50 | endocyclic and exocyclic | endocyclic and exocyclic | exocyclic (+ᵈ) and endocyclic | N.Mᶜ | Gong and Chen (2021) |

| Precursor | VOC conc. 1 | VOC conc. 2 | Reactor | Scavenger | T (K) | RH (%) | Mass conc. (M)[a] | Number conc. (N)[b] | M change trend | N change trend | Reference |
|---|---|---|---|---|---|---|---|---|---|---|---|
| | 1085 (15.2) | 900±10 (12200) | flow reactor | none | 298 | 3-62 (22-51) | 150; 200; 210 | N.M (23.8-55.3) | +[d] | −[e] (−[c]) | Li et al. (2019) |
| | 321±39 | 5786±203 | flow reactor | none | 298 | 0-60 | 980.9-2211.1 | 62.9-141.8 | +[d] (2 times) | +[d] (3 times) | this study |
| | 1000 | 1000 | flow reactor | cyclohexane | 295±2 | 0.02 and 32.5 | N.M[c] | N.M[c] | no effect | −[e] | Bonn et al. (2002) |
| Δ³-carene | 14.2 and 29.4 | 2300 | flow reactor | 2-butanol | 298±0.4 | <2-85 | 0.78-3.8 and 15.3-94; | 2.1-10.1and 19.8- 116.7 | +[d] | +[d] | Jonsson et al. (2006) |
| | 1111 | 900±10 | flow reactor | none | 298 | 3-62 | 75; 80; 90 | N.M | −[e] | −[e] | Li et al. (2019) |
| | 341±28 | 6257±140 | flow reactor | none | 298 | 0-60 | 346.0-198.5 | 19.4-11.1 | −[e] | no effect | this study |

[a] M means the change trend total particle mass concentration. [b] N means total particle number concentration. [c] N.M. means not mentioned. [d] Positive sign (+) means the mass or number concentration increases with RH. [e] Negative sign (−) means the mass or number concentration decreases with RH.

3. Also, I think the author should discuss the feasibility of extrapolating their flow tube results to the real environment. Limonene mixing ratios are at the sub-ppb level for forest and urban environments, thus the conditions applied in the current study seems far from atmospheric relevant. Could the authors explain the rational for using such high concentrations?

Response: To get enough SOA particles to analyze, many lab studies use VOC concentrations that are much higher than the ambient concentration. According to Table 2, the limonene concentration applied in previous studies was in the range of ~15-1000 ppb. The limonene concentration in this study (321 ppb) is within this range, but relatively high because we need to collect enough particles for off-line MS analysis with a small sampling flow through the OFR (0.9 L min$^{-1}$). To examine the feasibility of extrapolating our results to lower concentrations, we have performed a low-concentration limonene ozonolysis experiment. In this experiment, the limonene concentration was 20.5 ppb, ~16 times lower than previously used and close to the lower limit of the range applied in previous lab studies (i.e., ~15 ppb). According to the experimental results (Fig. S6), the number concentration of SOA formed from limonene ozonolysis increased by approximately 1.4 times under high RH, which is similar to the increase observed under high-loading conditions. The mass concentration increased by approximately 1.3 times at a precursor concentration of 20.5 ppb. The relatively small increase in mass concentration compared to the high-concentration conditions may be attributed to the less pronounced distribution of SVOCs at low mass concentrations. This result indicates that the enhancement effect on limonene SOA by high RH is still valid for low precursor concentrations. We have revised this at Page 18, Line 339-348: "To investigate the multi-generation reactions of limonene under low-concentration conditions, we conducted low-concentration limonene ozonolysis experiments, and the results are shown in Fig. S6. In these experiments, the limonene and O$_3$ concentrations were 20.5 ppb and 5.7 ppm, respectively. According to the experimental results, the number concentration of SOA formed from limonene ozonolysis increased by approximately 1.4 times under high RH, which is similar to the increase observed under high-loading conditions. The mass concentration increased by approximately 1.3 times at a precursor concentration of 20.5 ppb. The relatively small increase in mass concentration compared to the high-concentration conditions may be attributed to the less pronounced distribution of SVOCs at low mass concentrations. This result indicates that the enhancement effect on limonene SOA by high RH is still valid for low precursor concentrations."

[Figure]

**Figure S6.** The SOA formation of low-concentration limonene under low and high RH (a) mass concentration (b) number concentration (c) SOA yield (d) mean diameter.

4. Looking at Table 1, it seems that more $O_3$ is consumed in limonene experiments than in $\Delta^3$-carene experiments (if reported $O_3$ concentrations relates to measurement performed during the oxidation). To examine this, could the authors maybe report on the consumed $O_3$ (ppb) in all experiments (e.g. concentration before and after the OFR). In relation, have the authors considered the influence of OH-radicals as possible explanation for the differences in SOA formation from limonene and $\Delta^3$-carene? I wonder to what extent the resulting SOA from limonene and $\Delta^3$-carene can be ascribed to oxidation by OH vs $O_3$. I would expect that reaction with $O_3$ is the dominating oxidation pathway for limonene, whilst reactions with OH-radicals may be more significant in $\Delta^3$-carene experiments. Espec no such RH effect was observed for $O_3$+limonene. Consequently, although all experiments in the current study are conducted as dark ozonolysis of limonene and $\Delta^3$-carene, it might be important to address that this does not rule out the influence of other oxidation pathways (e.g. OH-radical reactions) which may be less effective at producing SOA compared to ozonolysis and which also could exhibit different response to RH. For instance, it may be that the $\Delta^3$-carene + OH reaction is unaffected (or enhanced relative to $\Delta^3$-carene + $O_3$ reactions) by RH (e.g. Bonn et al 2002) in contrast to the Limonene + $O_3$ reaction. The authors spend much effort on presenting and discussing the results related to the limonene experiments. However, in comparison, discussions on the $\Delta^3$-carene results seems lacking. In particular, results on the molecular analysis of the $\Delta^3$-carene SOA is lacking, e.g. comparison of mass spectrums recorded at different RH (such as in Figure 2), number and intensity proportion of the monomers, dimers, trimers and tetramers (as in Table S1).

Response: Following the Referee's suggestion, we have measured the $O_3$ consumptions, which are ~250 ppb for limonene experiments and ~100 ppb for $\Delta^3$-carene experiments. The text has been added in Page 4, Line 102-103: "Correspondingly, the $O_3$

consumption for limonene and Δ3-carene were ~250 ppb and ~100 ppb, respectively."

Molar OH radical yields were reported as 0.65±0.10 (Hantschke et al., 2021), 0.86±0.11 (Aschmann et al., 2002) and 0.56 to 0.59 (Wang et al., 2019) for $\Delta^3$-carene, while for limonene, the reported yields were 0.67±0.10 (Aschmann et al., 2002) and 0.76±0.06 (Herrmann et al., 2010). It seems that the OH radicals produced from limonene and $\Delta^3$-carene are quite similar within the range of uncertainties. Therefore, the increased ozone consumption by limonene is primarily attributed to the presence of its exocyclic double bond. We have also updated this in the revised manuscript (Page13, Line 254-261), "In such progress, we cannot rule out the possibility that relative humidity (RH) may influence the generation of other free radicals (Ma et al., 2009), thereby impacting the formation of secondary organic aerosols (SOA), such as, OH-radical reactions (Bonn et al., 2002; Fick et al., 2002). However, Molar OH radical yields were reported as 0.65±0.10 (Hantschke et al., 2021), 0.86±0.11 (Aschmann et al., 2002) and 0.56 to 0.59 (Wang et al., 2019) for $\Delta^3$-carene, while for limonene, the reported yields were 0.67±0.10 (Aschmann et al., 2002) and 0.76±0.06 (Herrmann et al., 2010). It seems that the OH radicals produced from limonene and $\Delta^3$-carene are quite similar within the range of uncertainties. Therefore, the increased ozone consumption by limonene is primarily attributed to the presence of its exocyclic double bond."

Additionally, we have included the mass spectra of SOA from $\Delta^3$-carene ozonolysis (Fig. S4) and the quantification of monomers, dimers, trimers, and tetramers of $\Delta^3$-carene, along with their corresponding number and intensity proportions in Table S2. The distribution of $\Delta^3$-carene SOA is similar to that of limonene-SOA , i.e., most of the SOA molecules are monomers (~70%) and dimers (~25%), while trimers and tetramers contribute to very small fractions (~2% and <1%). The corresponding discussion was changed in the revised manuscript (Page 11, Line 200-203): "Correspondingly, the distribution of $\Delta^3$-carene-SOA can be divided into four groups (Fig. S4), comparable to that of limonene-SOA. Most of the SOA molecules are monomers (~70%) and dimers (~25%), while trimers and tetramers contribute to smaller proportions (~2% and <1%, respectively) (Table S2)."

[Figure]

**Figure S4.** UPLC/ (−) ESI-Q-TOF-MS mass spectra of SOA from $\Delta^3$-carene ozonolysis. (a) MS under high and low RH conditions; (b) the identification of monomers under low RH condition.

**Table S2.** The number and intensity proportion of four groups for $\Delta^3$-carene

| Groups | Monomers | Dimers | Trimers | Tetramers |
|---|---|---|---|---|
| Number (L)[a] | 239 | 178 | 76 | 4 |
| Number (H)[b] | 216 | 151 | 26 | 1 |
| Intensity proportion (L)[a] | 69.8% | 28.6% | 1.6% | 0.5% |
| Intensity proportion (H)[b] | 72.5% | 26.9% | 2.0% | 0.2% |

[a]L means under low RH. [b]H means under high RH.

5. In relation, the observed increase in SOA mass in limonene experiments at elevated RH is proposed to arise from increased particle number concentration from nucleation promoted by low-volatile compounds such as dimers. To support this, the authors report 25 more dimers (187 vs 162) in limonene SOA formed at higher RH compared to low RH. This relatively small increase in LVOC species seems unlikely to account for the observed enhancements of SOA particle formation at high RH. At least the authors need to show that these extra dimers indeed contribute significantly to the formed SOA. Also, Could the authors please provide similar results from $\Delta^3$-carene experiments; i.e. how many dimers where found in $\Delta^3$-carene SOA and do the number of dimers change with changes in RH?

Response: To clarify the contribution of extra dimers, we conducted a reanalysis of the mass spectra for limonene SOA, specifically focusing on the dimers obtained under high RH conditions (Table S6). Among the 187 dimers observed, 54 (~19%) dimers were exclusively detected under high (RH) conditions. Note that some of the 162 dimers under low RH conditions were found under high RH conditions, so the number of newly formed dimers under high RH conditions (54) is larger than the absolute number difference (25). These particular dimers contribute to enhanced nucleation under high RH.

According to Table S2 (see our response above), the number of dimers in $\Delta^3$-carene SOA decreased under high RH conditions. As shown in Table S6, we observed 63 dimers exclusively under low humidity conditions, with a corresponding intensity of ~35%.

We have added the following text in the revised manuscript

Page 15, Line 275-277:

"As shown in Table S6, 54 out of the total 187 dimers were exclusively observed for limonene under high humidity conditions, contributing to a corresponding intensity of ~19%."

Page 18, Line 335-337:

"Correspondingly, the number and relative intensity of HOMs and dimers detected under high RH conditions are both lower than those under low RH conditions (Table S7). Furthermore, out of a total of 178 dimers, 63 dimers were exclusively identified under low RH conditions (Table S6)."

**Table S6.** Dimers: RH-dependent discoveries for limonene and $\Delta^3$-carene.

| 54 dimers exclusively detected under high RH (limonene) | | 63 dimers exclusively detected under low RH ($\Delta^3$-carene) | |
|---|---|---|---|
| Molecular formula | Absolute intensity (High RH) | Molecular formula | Absolute intensity (Low RH) |
| $C_{18}H_{26}O_4$ | $4.66\times10^2$ | $C_{17}H_{24}O_5$ | $1.59\times10^3$ |
| $C_{16}H_{20}O_6$ | $7.24\times10^2$ | $C_{10}H_{14}O_{11}$ | $3.90\times10^3$ |
| $C_{13}H_{18}O_9$ | $3.36\times10^2$ | $C_{14}H_{14}O_8$ | $4.02\times10^3$ |
| $C_{17}H_{22}O_6$ | $6.63\times10^3$ | $C_{20}H_{40}O_2$ | $4.60\times10^3$ |
| $C_{18}H_{26}O_5$ | $6.28\times10^2$ | $C_{12}H_{10}O_{10}$ | $4.00\times10^3$ |
| $C_{19}H_{32}O_4$ | $1.58\times10^3$ | $C_{13}H_{16}O_9$ | $8.34\times10^3$ |
| $C_{15}H_{18}O_8$ | $1.65\times10^3$ | $C_{19}H_{26}O_4$ | $4.96\times10^3$ |
| $C_{13}H_{12}O_{10}$ | $8.85\times10^3$ | $C_{17}H_{22}O_6$ | $1.05\times10^3$ |
| $C_{14}H_{20}O_9$ | $8.44\times10^2$ | $C_{13}H_{12}O_{10}$ | $5.46\times10^3$ |
| $C_{16}H_{28}O_7$ | $9.89\times10^3$ | $C_{13}H_{18}O_{10}$ | $4.68\times10^3$ |
| $C_{15}H_{26}O_8$ | $2.18\times10^3$ | $C_{15}H_{12}O_9$ | $4.22\times10^3$ |
| $C_{10}H_8O_{13}$ | $6.33\times10^3$ | $C_{10}H_{12}O_{13}$ | $5.00\times10^3$ |
| $C_{18}H_{24}O_6$ | $6.06\times10^2$ | $C_{22}H_{28}O_3$ | $8.88\times10^3$ |
| $C_{11}H_{14}O_{12}$ | $7.70\times10^2$ | $C_{19}H_{26}O_6$ | $1.54\times10^3$ |
| $C_{21}H_{22}O_4$ | $4.80\times10^3$ | $C_{16}H_{20}O_9$ | $1.64\times10^3$ |
| C20H34O4 | $2.53\times10^3$ | $C_{15}H_{18}O_{10}$ | $5.00\times10^3$ |
| $C_{23}H_{32}O_2$ | $2.12\times10^3$ | $C_{16}H_{22}O_9$ | $1.69\times10^3$ |
| $C_{18}H_{32}O_6$ | $3.68\times10^2$ | $C_{18}H_{22}O_8$ | $3.32\times10^3$ |
| $C_{17}H_{30}O_7$ | $7.46\times10^3$ | $C_{12}H_{16}O_{13}$ | $4.00\times10^3$ |
| $C_{14}H_{22}O_{10}$ | $4.04\times10^3$ | $C_{20}H_{32}O_6$ | $8.21\times10^3$ |
| $C_{21}H_{36}O_4$ | $1.36\times10^4$ | $C_{16}H_{18}O_{10}$ | $4.50\times10^3$ |
| $C_{17}H_{30}O_8$ | $4.68\times10^2$ | $C_{16}H_{20}O_{10}$ | $5.20\times10^3$ |
| $C_{12}H_{16}O_{13}$ | $2.43\times10^3$ | $C_{19}H_{24}O_8$ | $8.21\times10^3$ |
| $C_{11}H_{14}O_{14}$ | $4.46\times10^2$ | $C_{20}H_{28}O_7$ | $2.38\times10^3$ |
| $C_{18}H_{30}O_8$ | $4.46\times10^2$ | $C_{17}H_{20}O_{10}$ | $4.16\times10^3$ |
| $C_{16}H_{26}O_{10}$ | $7.44\times10^2$ | $C_{21}H_{36}O_6$ | $8.03\times10^3$ |
| $C_{17}H_{20}O_{10}$ | $2.12\times10^3$ | $C_{16}H_{26}O_{11}$ | $1.16\times10^3$ |
| $C_{16}H_{24}O_{11}$ | $1.48\times10^3$ | $C_{17}H_{26}O_{11}$ | $1.32\times10^3$ |
| $C_{20}H_{24}O_8$ | $3.96\times10^3$ | $C_{18}H_{18}O_{11}$ | $4.02\times10^3$ |
| $C_{17}H_{22}O_{11}$ | $2.48\times10^3$ | $C_{18}H_{22}O_{11}$ | $4.54\times10^3$ |
| $C_{21}H_{34}O_8$ | $1.28\times10^4$ | $C_{18}H_{26}O_{11}$ | $1.49\times10^3$ |
| $C_{13}H_{22}O_{15}$ | $4.06\times10^2$ | $C_{22}H_{28}O_8$ | $4.62\times10^3$ |
| $C_{19}H_{32}O_{10}$ | $5.30\times10^2$ | $C_{15}H_{18}O_{14}$ | $4.08\times10^3$ |

| Formula | Intensity | Formula | Intensity |
|---|---|---|---|
| $C_{22}H_{32}O_8$ | $5.90\times10^3$ | $C_{20}H_{32}O_{10}$ | $5.97\times10^3$ |
| $C_{20}H_{28}O_{10}$ | $1.53\times10^3$ | $C_{17}H_{22}O_{13}$ | $5.10\times10^3$ |
| $C_{18}H_{18}O_{13}$ | $4.49\times10^3$ | $C_{21}H_{28}O_{10}$ | $4.25\times10^3$ |
| $C_{19}H_{24}O_{12}$ | $1.49\times10^4$ | $C_{19}H_{22}O_{12}$ | $5.44\times10^3$ |
| $C_{19}H_{30}O_{12}$ | $6.10\times10^2$ | $C_{22}H_{34}O_9$ | $7.52\times10^3$ |
| $C_{15}H_{18}O_{16}$ | $1.14\times10^3$ | $C_{21}H_{34}O_{10}$ | $2.12\times10^3$ |
| $C_{23}H_{38}O_9$ | $4.34\times10^2$ | $C_{14}H_{24}O_{16}$ | $4.80\times10^3$ |
| $C_{32}H_{44}O_2$ | $8.96\times10^2$ | $C_{15}H_{22}O_{16}$ | $4.04\times10^3$ |
| $C_{21}H_{36}O_{11}$ | $3.74\times10^2$ | $C_{17}H_{30}O_{14}$ | $3.51\times10^3$ |
| $C_{14}H_{26}O_{17}$ | $1.00\times10^3$ | $C_{22}H_{36}O_{10}$ | $4.02\times10^3$ |
| $C_{20}H_{26}O_{13}$ | $1.26\times10^4$ | $C_{18}H_{24}O_{14}$ | $4.44\times10^3$ |
| $C_{22}H_{34}O_{11}$ | $1.92\times10^3$ | $C_{19}H_{28}O_{13}$ | $6.68\times10^3$ |
| $C_{20}H_{30}O_{13}$ | $9.36\times10^2$ | $C_{20}H_{22}O_{13}$ | $3.90\times10^3$ |
| $C_{18}H_{24}O_{15}$ | $2.05\times10^3$ | $C_{21}H_{26}O_{12}$ | $4.48\times10^3$ |
| $C_{21}H_{38}O_{12}$ | $9.16\times10^2$ | $C_{22}H_{30}O_{11}$ | $2.29\times10^3$ |
| $C_{24}H_{38}O_{10}$ | $3.78\times10^3$ | $C_{15}H_{24}O_{17}$ | $4.70\times10^3$ |
| $C_{16}H_{24}O_{17}$ | $1.26\times10^3$ | $C_{25}H_{38}O_9$ | $5.24\times10^3$ |
| $C_{21}H_{24}O_{14}$ | $4.80\times10^3$ | $C_{17}H_{26}O_{16}$ | $5.18\times10^3$ |
| $C_{20}H_{34}O_4$ | $4.98\times10^2$ | $C_{21}H_{26}O_{13}$ | $4.82\times10^3$ |
| $C_{18}H_{30}O_6$ | $2.74\times10^3$ | $C_{22}H_{30}O_{12}$ | $2.47\times10^3$ |
| $C_{18}H_{28}O_7$ | $1.53\times10^4$ | $C_{16}H_{24}O_{17}$ | $5.16\times10^3$ |
| | | $C_{17}H_{28}O_{16}$ | $6.58\times10^3$ |
| | | $C_{29}H_{44}O_6$ | $5.82\times10^3$ |
| | | $C_{17}H_{30}O_{16}$ | $2.06\times10^3$ |
| | | $C_{22}H_{38}O_{12}$ | $3.86\times10^3$ |
| | | $C_{16}H_{32}O_{17}$ | $7.04\times10^3$ |
| | | $C_{23}H_{30}O_{12}$ | $1.26\times10^3$ |
| | | $C_{24}H_{34}O_{11}$ | $6.82\times10^3$ |
| | | $C_{20}H_{30}O_{10}$ | $4.14\times10^3$ |
| | | $C_{20}H_{32}O_{11}$ | $3.41\times10^3$ |

6. What is the detection limit of the analytical method i.e. could the observation of the additional dimers (and HOMs) merely be due to higher filter mass loadings in high RH experiments. Excluding dimers and HOMs not found in low RH conditions, very little evidence is presented showing increased dimer and HOM formation at high RH. Also, despite more than 160 dimers found in LC-MS analysis of collected SOA, intensities are only reported for 5 dimers in limonene SOA and 7 dimers in $\Delta^3$-carene SOA (table S2 and S5). At least it would be beneficial to report how the intensities of these dimers change as a function of RH (not only high vs low RH). Particularly in Limonene experiments performed at 30, 40, 50 and 60 % RH where the particle number do not seem to changes significantly between experiments

Response: The absolute intensities of most monomers and dimers are relatively high ($>10^3$, see Table S3 and S7), which are much higher than the intensities of trimers and

tetramers that can still be detected by the LC-MS (Fig. 2). This indicates that the intensities of these dimers and HOMs are likely much higher than the detection limit.

In the original version of the manuscript, we specifically focused on reporting the products with proposed molecular structures, containing 5 dimers in limonene SOA and 7 dimers in $\Delta^3$-carene SOA. In the current version, we have added the molecular formulas of more dimers in Table S6. See our response above.

Due to the uncertainties related to filter collection and processing and the LC-MS itself, off-line analysis under every different RH is highly challenging. Thus, similar to the way applied in most of the previous off-line studies (Zhao et al., 2022; Li et al., 2020), we only collected and analyzed samples under the most extreme different conditions, i.e., the highest and lowest RH.

[Figure]

Fig. 2. UPLC/ (−) ESI-Q-TOF-MS mass spectra of SOA from limonene ozonolysis. (a) MS under high and low RH conditions; (b) the identification of monomers under high RH condition.

Other comments and suggestions:

7. Please add to Figure S3 time evolution of SOA size and mass concentration from all $\Delta^3$-carene/O$_3$ and limonene/O$_3$ experiments to validate the stable conditions of the OFR. Response: We have added the time evolution of SOA size and mass concentration from limonene/O$_3$ experiments in Fig. S3.

[Figure]

**Fig. S3**. Time evolution of SOA size (electromobility diameter) and mass concentration obtained from limonene/$O_3$ and $\Delta^3$-carene/$O_3$ experiments (Exp. 6 and Exp. 11).

8. Line 103-104: No description of materials are found in S2 (Figure?)

Response: S2 meant "Section S2. Materials" in the Supplement. We have changed "S2" into "Section S2" (Supplement, Page 2)

9. Line 259-260: Note that HOMs are not all considered low-volatile (see Kurtén et al. (2016), entitled "α-Pinene Autoxidation Products May Not Have Extremely Low Saturation Vapor Pressures Despite High O:C Ratios")

Response: We agree with the Referee that not all HOMs are considered low-volatile. We have changed this sentence into "Many HOMs have low volatilities (Donahue et al., 2011; Ehn et al., 2014)"

**Reference**

Aschmann, S. M., Arey, J., and Atkinson, R.: OH radical formation from the gas-phase reactions of O3 with a series of terpenes, Atmos. Environ., 36, 4347-4355, https://doi.org/10.1016/S1352-2310(02)00355-2, 2002.

Bonn, B., Schuster, G., and Moortgat, G. K.: Influence of water vapor on the process of new particle formation during monoterpene ozonolysis, J. Phys. Chem. A, 106, 2869-2881, https://doi.org/10.1021/jp012713p, 2002.

Donahue, N. M., Epstein, S. A., Pandis, S. N., and Robinson, A. L.: A two-dimensional volatility basis set: 1. organic-aerosol mixing thermodynamics, Atmos. Chem. Phys., 11, 3303-3318, 10.5194/acp-11-3303-2011, 2011.

Ehn, M., Thornton, J. A., Kleist, E., Sipila, M., Junninen, H., Pullinen, I., Springer, M., Rubach, F., Tillmann, R., Lee, B., Lopez-Hilfiker, F., Andres, S., Acir, I.-H., Rissanen, M., Jokinen, T., Schobesberger, S., Kangasluoma, J., Kontkanen, J., Nieminen, T., Kurten, T., Nielsen, L. B., Jorgensen, S., Kjaergaard, H. G., Canagaratna, M., Dal Maso, M., Berndt, T., Petaja, T., Wahner, A., Kerminen, V.-M., Kulmala, M., Worsnop, D. R., Wildt, J., and Mentel, T. F.: A large source of low-volatility secondary organic aerosol, NATURE, 506, 476-+, 10.1038/nature13032, 2014.

Fick, J., Pommer, L., Andersson, B., and Nilsson, C.: A study of the gas-phase ozonolysis of terpenes: the impact of radicals formed during the reaction, Atmos. Environ., 36, 3299-3308, https://doi.org/10.1016/s1352-2310(02)00291-1, 2002.

Gong, Y. and Chen, Z.: Quantification of the role of stabilized Criegee intermediates in the formation of aerosols in limonene ozonolysis, Atmos. Chem. Phys., 21, 813-829, https://doi.org/10.5194/acp-21-813-2021, 2021.

Hantschke, L., Novelli, A., Bohn, B., Cho, C., Reimer, D., Rohrer, F., Tillmann, R., Glowania, M., Hofzumahaus, A., Kiendler-Scharr, A., Wahner, A., and Fuchs, H.: Atmospheric photooxidation and ozonolysis of Δ3-carene and 3-caronaldehyde: rate constants and product yields, Atmos. Chem. Phys., 21, 12665-12685, 10.5194/acp-21-12665-2021, 2021.

Herrmann, F., Winterhalter, R., Moortgat, G. K., and Williams, J.: Hydroxyl radical (OH) yields from the ozonolysis of both double bonds for five monoterpenes, Atmos. Environ., 44, 3458-3464, https://doi.org/10.1016/j.atmosenv.2010.05.011, 2010.

Jonsson, A. M., Hallquist, M., and Ljungstrom, E.: Impact of humidity on the ozone initiated oxidation of limonene, Delta(3)-carene, and alpha-pinene, Environ. Sci. Technol., 40, 188-194, https://doi.org/10.1021/es051163w, 2006.

Li, J., Wang, W., Li, K., Zhang, W., Peng, C., Zhou, L., Shi, B., Chen, Y., Liu, M., Li, H., and Ge, M.: Temperature effects on optical properties and chemical composition of secondary organic aerosol derived from n-dodecane, Atmos. Chem. Phys., 20, 8123-8137, 10.5194/acp-20-8123-2020, 2020.

Liu, Q., Liggio, J., Breznan, D., Thomson, E. M., Kumarathasan, P., Vincent, R., Li, K., and Li, S.-M.: Oxidative and Toxicological Evolution of Engineered Nanoparticles with Atmospherically Relevant Coatings, Environ. Sci. Technol., 53, 3058-3066,

10.1021/acs.est.8b06879, 2019.

Liu, Y., Liggio, J., Harner, T., Jantunen, L., Shoeib, M., and Li, S.-M.: Heterogeneous OH Initiated Oxidation: A Possible Explanation for the Persistence of Organophosphate Flame Retardants in Air, Environ. Sci. Technol., 48, 1041-1048, 10.1021/es404515k, 2014.

Ma, Y., Porter, R. A., Chappell, D., Russell, A. T., and Marston, G.: Mechanisms for the formation of organic acids in the gas-phase ozonolysis of 3-carene, Phys. Chem. Chem. Phys., 11, 4184-4197, 10.1039/b818750a, 2009.

Wang, L., Liu, Y., and Wang, L.: Ozonolysis of 3-carene in the atmosphere. Formation mechanism of hydroxyl radical and secondary ozonides, Phys. Chem. Chem. Phys., 21, 8081-8091, 10.1039/c8cp07195k, 2019.

Zhao, Y., Yao, M., Wang, Y., Li, Z., Wang, S., Li, C., and Xiao, H.: Acylperoxy Radicals as Key Intermediates in the Formation of Dimeric Compounds in α-Pinene Secondary Organic Aerosol, Environ. Sci. Technol., 56, 14249-14261, 10.1021/acs.est.2c02090, 2022.

---

## Author Comment (AC3)

**Response to the comments of Anonymous Referee #3**

General comments:

The manuscript by Zhang et al. investigated the effect of relative humidity (RH) on the formation of secondary organic aerosols (SOA) from structurally distinct monoterpenes from a molecular-level perspective. They observed a significant difference between limonene and $\Delta3$-carene SOA formation on RH dependence. Further, they proposed potential chemical reaction mechanisms and pathways based on mass spectrometry analysis to explain the observed increase in limonene SOA and the decrease in $\Delta3$-carene SOA. They suggested that the exocyclic double bond in limonene plays an important role in multi-generation reactions, contributing to the formation of lower volatile compounds under high RH conditions. Compared to many previous studies on the RH effects of SOA formation from monoterpenes, this study provides important insights into the multi-generation reactions that drive SOA formation by applying high-resolution MS analysis. The findings of this manuscript have significant implications for a better understanding of the mechanism of monoterpene oxidation reactions and the generation of secondary organic aerosols. This manuscript is well-written and I recommend publication in Atmospheric Chemistry and Physics after addressing the following minor concerns.

Response: We thank the Referee for the valuable comments and suggestions regarding our manuscript. We have made revisions to address these issues and believe that the updated version significantly improves the quality and meet the standard of ACP. The major revisions are as follows:

1. We have conducted low precursor concentration experiments to investigate the influence of SOA loading on multi-generation reactions of limonene.

2. We have supplemented the distribution of mass spectrum of $\Delta^3$-carene to provide a more comprehensive understanding for the increase of number concentration under low RH.

The responses are listed below in blue color text and the associated revisions to the manuscript are shown in red color text.

Specific comments:

1. The authors used high precursor concentrations in the experiments. It would be more convincing if similar results can be found with lower precursor concentrations.

Response: We have added the low-concentration limonene ozonolysis experiment (Fig. S6). The details are updated in the revised manuscript at Page 18, Line 339-348.

"To investigate the multi-generation reactions of limonene under low-concentration conditions, we conducted low-concentration limonene ozonolysis experiments, and the results are shown in Figure S6. In these experiments, the limonene and $O_3$ concentrations were 20.5 ppb and 5.7 ppm, respectively. According to the experimental results, the number concentration of SOA formed from limonene ozonolysis increased by approximately 1.4 times under high RH, which is similar to the increase observed under high-loading conditions. The mass concentration increased by

approximately 1.3 times at a precursor concentration of 20.5 ppb. The relatively small increase in mass concentration compared to the high-concentration conditions may be attributed to the less pronounced distribution of SVOCs at low mass concentrations. This result indicates that the enhancement effect on limonene SOA by high RH is still valid for low precursor concentrations."

[Figure]

**Figure S6**. The SOA formation of low-concentration limonene under low and high RH (a) mass concentration (b) number concentration (c) SOA yield (d) mean diameter.

2. The authors have extensively described the mass spectrometry analysis of limonene, while only briefly providing the distribution of high oxygenated compounds and dimers in $\Delta^3$-carene. It would be beneficial to include additional analysis of $\Delta$3-carene mass spectrometry.

Response: Like limonene, we have analyzed the number and intensity proportion of four groups for $\Delta^3$-carene. The details have given in Table S2 in the Supplement, and the distribution is similar to that of limonene-SOA, i.e., most of the SOA molecules are monomers (~70%) (Fig. 2b) and dimers (~25%), while trimers and tetramers contribute to very small fractions (~2% and <1%). The corresponding discussion was changed in the revised manuscript (Page 11, Line 200-203): "Correspondingly, the distribution of $\Delta^3$-carene-SOA can be divided into four groups (Fig. S4), comparable to that of limonene-SOA. Most of the SOA molecules are monomers (~70%) and dimers (~25%), while trimers and tetramers contribute to smaller proportions (~2% and <1%, respectively) (Table S2)."

Table S2. The number and intensity proportion of four groups for $\Delta^3$-carene.

| Groups | Monomers | Dimers | Trimers | Tetramers |
|---|---|---|---|---|

| | | | | |
|---|---|---|---|---|
| **Number (L)a** | 239 | 178 | 76 | 4 |
| **Number (H)b** | 216 | 151 | 26 | 1 |
| **Intensity proportion (L)a** | 69.8% | 28.6% | 1.6% | 0.5% |
| **Intensity proportion (H)b** | 72.5% | 26.9% | 2.0% | 0.2% |

a L means under low RH. b H means under high RH.

3. Method: what is the temperature ramp program in liquid chromatography?

Response: The temperature ramp program was: 0–3min with 0%–3% phase B, 3–25min with 3%–50% phase B, 25–43min with 50%–90% phase B, 43–48 min with 90%–3% phase B, 48–60min with 3% phase B. The corresponding content has been added in the revised manuscript at Page 6, Line 138-140.

R4: Page 5, Line116: "limonene-" should be "limonene-SOA".

Response: Revised.

R5: Page 7, Line 156: the authors have pointed out that the OH scavenger will produce additional products which may influence the reactions of target precursors, so what about the 2-butanol and cyclohexane discussed in this article?

Response: Though there is no difference between 2-butanol and cyclohexane in the scavenging ability of OH radical, 2-butanol will produce more $HO_2\cdot$ than cyclohexane and, consequently, $R\cdot$ will react with $HO_2\cdot$ to produce more hydroxyl acids and hydroxyl per-acid products, most of which have low volatility, thus high partitioning into the particle phase. The corresponding content has been revised in Page 7, Line 161-165: "For example, there is no difference between 2-butanol and cyclohexane in the scavenging ability of OH radical, though 2-butanol will produce more $HO_2\cdot$ than cyclohexane and, consequently, $R\cdot$ will react with $HO_2\cdot$ to produce more hydroxyl acids and hydroxyl per-acid products, most of which have low volatility and, thus high partitioning into the particle phase."

R6: Page12, Line 238-239: specify the condition of the increase and decrease of $C_9H_{14}O_3$ and $C_{10}H_{16}O_2$, respectively.

Response: In order to avoid ambiguity, we have revised the sentence to "This mechanism can well explain the decrease in the relative intensity of $C_{10}H_{16}O_2$ from high RH to low RH and the increase in the relative intensity of $C_9H_{14}O_3$ from low RH to high RH (Table S3)".

R7: Supplement, Page 11: there were two (K) in Table S6.

Response: Revised.

---

## Author Response (AR2)

Response to Reviewer #1

The authors have put in good efforts to improve on the manuscript based on the reviews. I am still hesitant to believe that the effects they observe are purely due to the sCI reactions they propose. However, I can accept that this manuscript will have value for the wider community. My only requirement at this stage is that the authors make it very clear in the manuscript that the reactions they suggest are indeed suggestions, and that other factors might also be of importance.

Response: we thank the referee for providing the feedback on our revised manuscript, we have fully considered the comments, responded to these comments below in blue text and made the associated revisions to the manuscript as shown in red text. The response and changes are listed below.

Despite a vast literature on chemical reactions taking place in the particle phase, the authors currently dismiss such reactions by stating that they see an equivalent increase in particle number and in mass, which to them suggests that the change happens in the gas phase. This is a very speculative conclusion, as aerosol dynamics of small clusters and particles are very complex. One could also ask the authors to explain why they do not see any change in the particle phase chemistry as a function of RH.

In order to avoid making a very strong claim that large changes happen in the gas phase chemistry, and no changes happen in the condensed phase chemistry, my strong recommendation to the authors is to highlight both in the abstract, discussions, and conclusions, that they cannot rule out that other reactions may also be taking place, for example in the particle phase, and that the presented findings are their interpretations. Phrases like "This hypothesis is further proved..." in the abstract and conclusions should thus be reformulated, e.g. by using "supported" instead of "proved", in addition to adding some discussion that other potential mechanisms could also be taking place.

Response: We agree with the referee that reactions in particle phase may also play a role. Following the Referee's suggestion, we have added and changed some text in the revised manuscript.

Page 1, Line 17-19: Although the complex processes in the particle phase may play a role, we primarily attribute it to the water-influenced reactions after ozone attack on the exocyclic double bond of limonene, which leads to the increment of lower volatile organic compounds under high RH condition.

Page 1, Line 20-21: This hypothesis is further supported by the SOA yield enhancement of β-caryophyllene, a sesquiterpene that also has an exocyclic double bond.

Page 13, Line 259-260: Therefore, the increased ozone consumption by limonene seems primarily attributed to the presence of its exocyclic double bond.

Page 15, Line 285-286: Overall, the promoted dimer and HOM formation may greatly enhance the new particle number concentration under high RH condition (Fig. 6).

Page 17, Line 314-315: Overall, it is likely that the different fate and partitioning of SVOCs largely enhance the amount of SVOCs in the particle phase (Fig. 6).

Page 17, Line 317-319: While our study highlights significant changes in gas-phase chemistry, we cannot exclude the possibility of concurrent reactions occurring in the condensed phase.

Page 18, Line 348-349: This result suggests that the enhancement effect on limonene SOA by high RH is still valid for low precursor concentrations.

Page 19, Line 373-376: Moreover, this hypothesis is supported by a similar behavior of the ozonolysis of β-caryophyllene (sesquiterpene with an exocyclic double bond) in SOA enhancement under high RH condition. However, since aerosol dynamics of small clusters and particles are very complex, we do not rule out a series of reactions that may occur in the particle phase.

Minor comment: The argument that sCI chemistry produces carbonyls and that these explain changes in SOA yields still seems questionable, as gas phase chemistry in general produces carbonyls. Also in the mechanism figures in this paper, carbonyls are the major functionalities in almost all reaction pathways.

Response: We agree with the referee that carbonyls can be formed in other gas-phase reactions, e.g., during the production of OH and oxygen addition process as illustrated in Figure 3. For limonene, the presence of exocyclic double bonds allows these carbonyl compounds to produce compounds with multiple carbonyl groups, further lower its volatility and enhancing the SOA formation. Furthermore, Gong and Chen (2021) indicated that the SOA formation potential of SCIs under high-humidity conditions was nearly double that under dry and low-humidity conditions. Since carbonyl compounds are the major products in the SCI channel (Leungsakul et al., 2005), it is reasonable to infer that the carbonyl compounds generated via SCI under high humidity conditions significantly contribute to SOA formation.

Furthermore, the oligomerization of these carbonyls generates more dimers including hemiacetal (or acetal) formation and aldol condensation, and the intensity of dimer formation originating from multi-carbonyls are enhanced under high RH conditions to low RH conditions (Table S6) (Jang et al., 2003). The corresponding revision in the manuscript is in Page 15, Line 274-275: "As shown in Table S6, the intensity of dimers generating from multi-carbonyls under high RH is higher than that under low RH."

**Table S6.** The intensity of dimers from multi-carbonyls under high RH and low RH

| Molecular formula | Absolute intensity (Low RH) | Relative intensity (Low RH) | Absolute intensity (High RH) | Relative intensity (High RH) |
|---|---|---|---|---|
| $C_{19}H_{28}O_5$ | $3.60\times10^2$ | $6.87\times10^{-5}$ | $9.28\times10^3$ | $2.06\times10^{-3}$ |
| $C_{19}H_{28}O_7$ | $5.00\times10^3$ | $9.5\times10^{-4}$ | $2.73\times10^4$ | $6.08\times10^{-3}$ |
| $C_{19}H_{28}O_6$ | $1.78\times10^3$ | $3.40\times10^{-4}$ | $1.39\times10^4$ | $3.10\times10^{-3}$ |
| $C_{18}H_{28}O_6$ | $3.19\times10^3$ | $6.08\times10^{-4}$ | $4.30\times10^3$ | $9.56\times10^{-4}$ |
| $C_{18}H_{24}O_6$ | | | $6.06\times10^2$ | $1.35\times10^{-4}$ |
| $C_{18}H_{26}O_5$ | | | $6.28\times10^2$ | $1.40\times10^{-4}$ |

**Reference:**

Gong, Y. and Chen, Z.: Quantification of the role of stabilized Criegee intermediates in the formation of aerosols in limonene ozonolysis, Atmos. Chem. Phys., 21, 813-829, https://doi.org/10.5194/acp-21-813-2021, 2021.

Jang, M. S., Carroll, B., Chandramouli, B., and Kamens, R. M.: Particle growth by acid-catalyzed heterogeneous reactions of organic carbonyls on preexisting aerosols, Environ. Sci. Technol., 37, 3828-3837, 10.1021/es021005u, 2003.

Leungsakul, S., Jaoui, M., and Kamens, R. M.: Kinetic Mechanism for Predicting Secondary Organic Aerosol Formation from the Reaction of d-Limonene with Ozone, Environ. Sci. Technol., 39, 9583-9594, https://doi.org/10.1021/es0492687, 2005.